# Clinical and pharmacokinetic/dynamic outcomes of prolonged infusions of beta-lactam antimicrobials: An overview of systematic reviews

**Pierre Thabet[1], Anchal Joshi[2], Erika MacDonald[1], Brian Hutton[3,4], Wei Cheng[3], Adrienne Stevens[5], Salmaan Kanji** [1,3,4]*

**1** The Ottawa Hospital, Ottawa, Ontario, Canada, **2** University of Waterloo, Waterloo, Ontario, Canada, **3** Ottawa Hospital Research Institute, Ottawa, Ontario, Canada, **4** University of Ottawa School of Epidemiology and Public Health, Ottawa, Canada, **5** McMaster University, Hamilton, Ontario, Canada

* skanji@toh.ca

## Abstract

### Objective

This overview of reviews aims to map and compare of objectives, methods, and findings of existing systematic reviews to develop a greater understanding of the information available about prolonged beta-lactam infusions in hospitalized patients with infection.

### Design

Overview of systematic reviews.

### Data sources

Medline, Embase, PROSPERO and the Cochrane Library were systematically searched from January, 1990 to June, 2019 using a peer reviewed search strategy. Grey literature was also searched for relevant reviews.

### Eligibility criteria for selecting reviews

Systematic reviews were sought that compared two or more infusion strategies for intravenous beta-lactam antimicrobials and report clinical cure or mortality. Populations of included reviews were restricted to hospitalized patients with infection, without restrictions on age, infection type, or disease.

### Data extraction and analysis

Abstract screening, data extraction, quality and risk of bias assessment were conducted by two independent reviewers. Overlap between reviews was assessed using a modified corrected covered area. Overview findings are reported in accordance with Cochrane's recommendation for overview conduct. Clinical outcomes extracted included survival, clinical cure, treatment failure, microbiological cure, length of stay, adverse events, cost, and emergence of resistance.

**Data Availability Statement:** All relevant data are within the manuscript and its Supporting information files.

**Funding:** The authors received no specific funding for this work.

**Competing interests:** The authors have declared that no competing interests exist.

## Results

The search strategy identified 3327 unique citations from which 21 eligible reviews were included. Reviews varied by population, intervention and outcomes studied. Between reviews, overlap of primary studies was generally high, methodologic quality generally low and risk of bias variable. Nine of 14 reviews that quantitatively evaluated mortality and clinical cure identified a benefit with prolonged infusions of beta lactams when compared with intermittent infusions. Evidence of mortality and clinical cure benefit was greater among critically ill patients when compared to less sick patients and lower in randomized controlled trials when compared with observational studies.

## Conclusions

Findings from our review demonstrate a consistent and reproducible lack of harm with prolonged infusions of beta-lactam antibiotics with variability in effect size and significance of benefits. Despite 21 systematic reviews addressing prolonged infusions of beta-lactams, this overview supports the continued need for a definitive systematic review given variability in populations, interventions and outcomes in the current systematic reviews. Subsequent systematic reviews should have more rigorous and transparent methods, only include RCTs and evaluate the proposed benefits found in various subgroup-analyses—i.e. high risk of mortality.

## Trial registration

Prospero registry, CRD42019117118.

## Introduction

Despite antimicrobial therapy, many patients experience negative infection related outcomes. This is particularly true of patients with sepsis and the critically ill [1, 2]. Advancements in supportive care and bundled care initiatives such as the *Surviving Sepsis Campaign* have minimal impact on sepsis-associated morbidity and mortality [3, 4]. Meanwhile, antimicrobial resistance has increased at a staggering pace. Current guidelines for antimicrobial use focus on what is administered and when it is administered without considering how it is administered [5]. The method of antibiotic administration, particularly for beta-lactam antibiotics, can have a substantial impact on patient outcomes [6]. Accordingly, there is an urgent need to establish the optimal method of beta-lactam antimicrobial administration in specific patient populations.

The greatest predictor of antimicrobial-related treatment failure is inadequate dosing that results in sub-therapeutic serum drug concentrations [7]. Up to 50% of septic patients are inadequately dosed due to extremely variable pharmacokinetics (PK), such as drug clearance, metabolism and volume of distribution differences [8–10]. Typically, pharmacodynamic (PD) targets from 40–70% time above the MIC ($f\text{T}_{>\text{MIC}}$) are associated with improved outcomes from animal and human studies for beta-lactam antimicrobials depending on the sub-class of beta-lactam (i.e., penicillins, cephalosporins and carbapenems) and the organism causing the infection [11, 12]. Growing concern as to generalizability of this target has been expressed, and there is a mounting evidence supporting a target of 100% time above the MIC for critically ill patients and those with life threatening infections with resistant organisms [13, 14]. PD models

support increasing the infusion time of beta-lactams to improve the probability of attaining adequate serum antimicrobial concentrations, and clinical trials have demonstrated that greater survival can be achieved in critically ill patients receiving prolonged infusions of beta-lactam antimicrobials [15, 16]. Despite this, most beta-lactam antimicrobials are administered as intermittent boluses, with unpredictable PK contributing to subtherapeutic serum concentrations and antimicrobial-related treatment failure.

While antimicrobial stewardship strategies that encourage appropriate use of antimicrobials have had some success in slowing the development of resistance, a greater understanding of methods to optimize the use of antimicrobials is necessary [17]. Altering the method of administration of beta-lactams to achieve optimal pharmacodynamics represents a low-cost, low-risk change in practice that can slow the development of resistance and improve the outcomes of patients with sepsis and septic shock [18–21].

Various methods of administration of intravenous beta-lactam antimicrobials exist that are potentially associated with different PD target attainment rates and patient outcomes. Intermittent doses are generally given over 30 or 60 minutes, one to six times per day depending on the drug. Extended infusions are administered over 3–4 hours per dose, while for continuous infusions, the entire daily dose is infused over 24 hours.

The literature evaluating prolonged infusion of beta-lactam antibiotics has been polarizing due to both positive and equivocal studies, of which there are over 60. Systematic reviews exist that evaluate the clinical impact of antimicrobial administration method, but the messaging is again inconsistent due to variable populations, outcomes and quality of the reviews [22–36]. Conducting an overview of reviews is a suitable method for describing the evidence across reviews and investigating the source of discrepancies. An overview of reviews uses systematic methods to identify existing systematic reviews with similar research questions for the purpose of mapping and comparing objectives, methods, and findings to develop a greater understanding of the information available. Overviews of reviews are considered suitable when multiple systematic reviews of the same topic exist but 1)the interventions are different for the same condition or population, 2) they address different approaches to the same intervention in the same population, 3) they address the same intervention but for different populations or conditions, 4) they address adverse events among different populations or 5) they address the different outcomes for the same intervention and population. In this case there are multiple systematic reviews on the topic of prolonged infusions of beta lactam antibiotics. The populations in these reviews vary from neonates to adults and the spectrum of infections vary in site, organism and severity. The interventions studied are all prolonged infusions of a beta-lactam antibiotic, but the specific agent varies within and between sub-classes of beta-lactam antibiotics as does the method of prolonging the infusion (i.e., continuous versus extended). Finally, although mortality is the most commonly investigated outcome, other important clinical and PK/PD outcomes are variably reported. This overview of reviews addresses the following question: "In hospitalized patients with infection treated with intravenous beta-lactam antibiotics, what is the evidence from existing systematic reviews comparing prolonged with intermittent infusions of beta-lactam antibiotics for clinical outcomes including mortality, clinical and microbiological cure, length of stay, adverse events, cost and PK/PD target attainment?"

## Methods

This overview of reviews was guided by the Cochrane Handbook [37] on methods for overviews of reviews, the PRISMA guidelines for reporting of systematic reviews [38, 39] and PRIO-harms checklist [40]. The protocol was prospectively registered in PROSPERO (CRD42019117118). Minimal deviations from the protocol were incurred; review methods

modified in accordance to the newly revised Cochrane handbook and risk of bias and quality assessment were conducted by one reviewer and verified by a second reviewer instead of being performed in duplicate (due to limited availability of resources).

## Searching the literature

A systematic search strategy for Medline, Embase, and the Cochrane Library, was created by an experienced information specialist and reviewed by a second information specialist using the PRESS criteria (S1 Fig) [41]. PROSPERO was searched for ongoing or unreported relevant reviews and to cross reference database searches. Grey literature was searched in accordance with the *Grey Matters* Checklist (limited to Google, Google Scholar, the Agency for Healthcare Research Quality, Cochrane Library of Systematic Reviews using the terms beta-lactam, penicillin, carbapenem, cephalosporin, continuous, intermittent and prolonged infusions) and bibliographies of included reviews were scanned for additional reviews [42]. The search focused on reviews from January, 1990 to June, 2020, as a scoping exercise revealed no relevant systematic reviews prior to 1990.

## Study selection criteria

Eligible systematic reviews for this overview were included based upon inclusion of the following features: (a) a systematic search of the literature with a clearly defined research question and eligibility criteria; and (b) a quantitative or narrative synthesis of extracted data. Populations were restricted to hospitalized patients with infection, without restrictions on age, infection type, or disease. Eligible reviews were required to have compared two or more infusion strategies (intermittent, extended or continuous) for administration of intravenous beta-lactam antimicrobials, with data available for clinical cure (no restrictions on definition) or mortality/survival. For reviews that reported treatment failure instead of clinical cure, the inverse was calculated and reported as treatment success or clinical cure. Secondary clinical outcomes of interest included treatment failure, microbiological cure, length of stay, adverse events, cost and emergence of resistance. Variable definitions of clinical outcomes were expected and collected but were not pre-specified during development of the methods for this overview. PK/PD outcomes of interest were probability of target attainment and $f$T>MIC.

## Processes of study selection, data extraction and analysis

Two reviewers (AJ and PT) independently screened search results by title and abstract. Citations identified as potentially relevant by at least one reviewer underwent full text-review by both reviewers independently. Discrepancies in full text screening were resolved by a third reviewer (SK). Prespecified review characteristics and outcomes of interest, as defined in their respective reviews, were extracted by two reviewers (AJ and PT), with discrepancies resolved by consensus or a third reviewer (SK). Quality assessment of each included review was conducted using the AMSTAR-2 tool (Assessing the Methodological Quality of Systematic Reviews), and risk of bias of assessments were performed using with the ROBIS tool (Risk of Bias in Systematic Reviews); all assessments were performed by one reviewer (PT) and verified by another (SK), with disagreements resolved via consensus discussion [43, 44]. Domains from AMSTAR-2 deemed to be critical to this overview included 1) protocol registration prior to starting the review, 2) adequacy of the literature search, 3) justification for study exclusions, 4) risk of bias assessment at the study level, 5) appropriateness of methods for meta-analysis, 6) consideration of risk of bias upon interpretation of results and 7) assessment of publication bias. Risk of bias at the study level and outcome level was also extracted from reviews as reported including the tools used for assessment.

Overlap of included studies was assessed using the corrected covered area (CCA) [45]. CCA calculations followed Pieper et al's protocol according to $CCA = \frac{N-r}{rc-r}$, where N is the number of included publications (including double counting), r is the number of primary publications, and c is the number of reviews. Pre-determined overlap thresholds were used for interpretation of overlap (0–5%—slight, 6–10%—moderate, 11–15%—high, >15%—very high) [45]. For each outcome, a citation matrix and pairwise CCA tables were created in addition to the outcome level CCA calculations to address overlap. Overlap is presented visually as per recommendations from Pérez-Bracchiglione et al. at the 2019 Cochrane Colloquium [46].

Overview findings are reported descriptively by outcome, following Cochrane's recommendation for overview conduct and the PRIO-harms checklist (S2 Fig) [37, 40]. Tables of study characteristics were created at the review level and at the outcome level for this evaluation. For outcomes where meta-analysis was conducted by more than two reviews, forest plots without cumulative statistics were created for visual comparison.

## Results

We identified 3,407 unique records from searching electronic databases, as well as 3 additional records through hand searching of references and grey literature. 78 articles met our pre-specified criteria for full text assessment. After inspection of full texts, 21 reviews were included in this overview of systematic reviews (Fig 1) [16, 22–33, 36, 47–53]. Table 1 provides an overview of the review characteristics.

### Review populations and interventions studied

The 21 reviews varied by population, intervention, and outcomes. Ten reviews did not specify the patient population studied [22, 26–28, 30, 31, 36, 48, 49, 53]. Nine reviews studied patients with acute illness, critical illness or sepsis [16, 23–25, 29, 32, 33, 47, 52]. The remaining 2 studied patients with respiratory infections [50, 51]. 9 reviews studied prolonged infusions of all beta-lactams [16, 23, 24, 26–28, 48, 50, 52]. 4 reviews studied individual classes or sub-classes of beta-lactams—cephalosporins, anti-pseudomonal beta-lactams, carbapenems and piperacillin/tazobactam [29, 36, 47, 49]. The remaining 8 reviews studied individual antimicrobials, with 7 studying piperacillin/tazobactam, 1 meropenem and 1 cefepime [22, 25, 28, 30–33, 53]. 3 reviews exclusively compared continuous infusions of beta-lactams with intermittent infusions [16, 23, 52], whereas the remaining 18 reviews compared all prolonged infusions with intermittent infusions. 15 reviews quantitatively analyzed outcomes of interest [16, 23–27, 29–32, 36, 47–50], whereas 6 narratively describe outcomes of interest [22, 28, 33, 51–53]. Overlap of primary studies and proportion of unique primary studies were highly variable depending on the outcome assessed, ranging from 4.8% to 13.2% and 36% to 78% respectively.

### Review quality and risk of bias

Based upon findings from AMSTAR-2 evaluations that were performed to evaluate the methodological quality of included systematic reviews, almost all reviews were deemed to be of critically low or low quality. (Table 1, full assessments provided in S1 Table). Only two reviews were deemed to be of moderate quality [27, 51]. Ten reviews were deemed to have high risk of bias (Fig 2) [23, 24, 26, 28, 33, 36, 47–49, 52]. Acknowledging the overlap (and inverse relationship) between AMSTAR-2 and ROBIS discordant judgements between the two tools were primarily related to items unique to each tool.

Sixteen of 21 reviews provide study-level assessments of risk of bias using a variety of tools. Authors of one review [29] indicated they used GRADE to assess risk of bias for a subset of studies. One review stated that they used the Cochrane risk of bias tool but did not provide

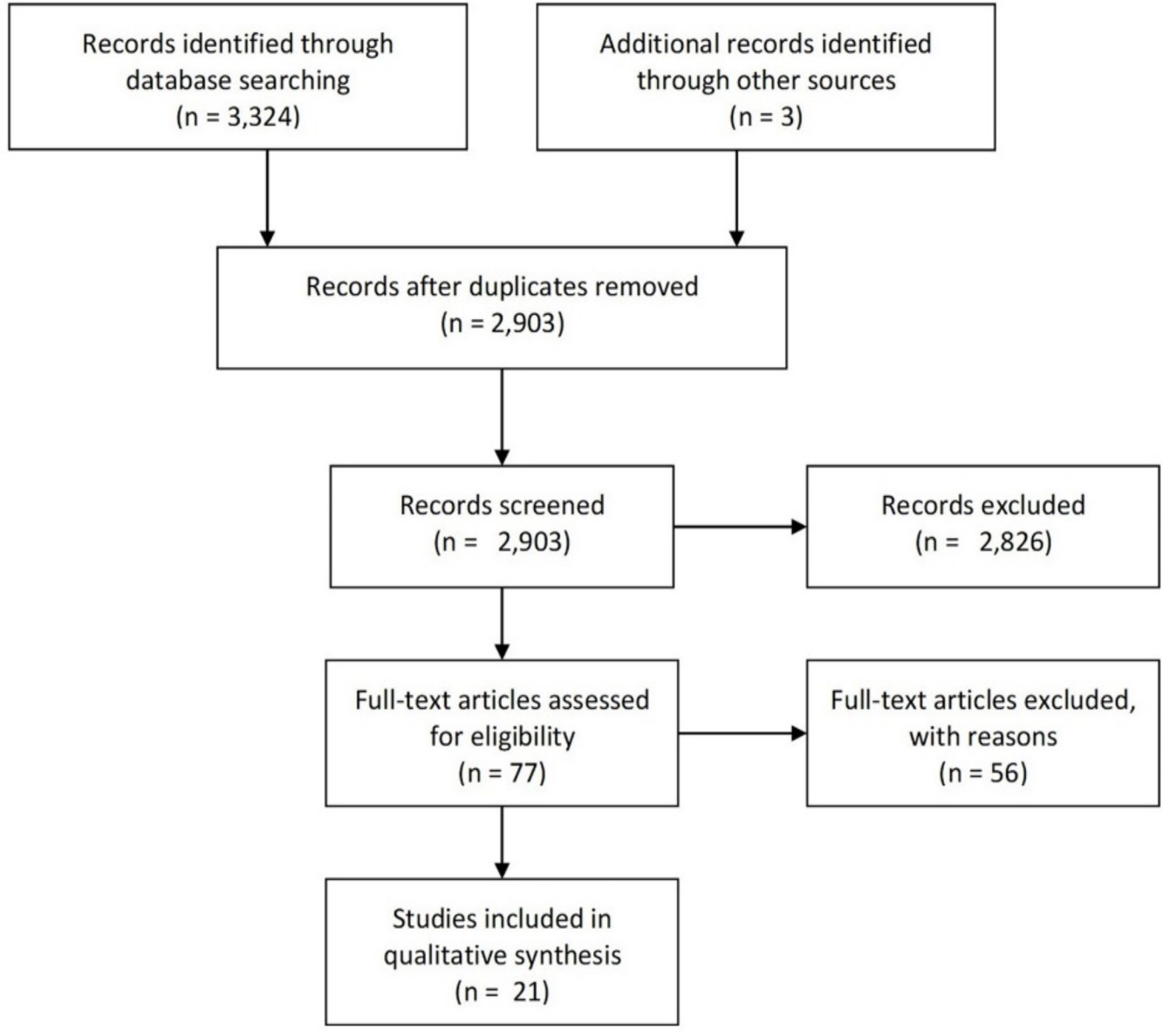

**Fig 1. Study flow diagram.** Results of systematic review identification, screening, and inclusion.

information describing their evaluation [31]. None of the reviews provided a final assessment of risk of bias by outcome. Remaining reviews that included randomized controlled trials evaluated risk of bias using the Jadad scale [24, 32, 48, 50] or their own criteria [23, 28, 33, 47]. In relation to non-randomized studies, the majority of reviews used the Newcastle-Ottawa scale [24, 30, 31, 50], while three reviews used their own criteria [23, 28, 47]. Seven reviews did not report that they evaluated risk of bias at the study level at all [22, 26, 28, 36, 49, 52, 53]. Given the variable reporting and poor adherence to practice standards (i.e., GRADE framework), it was impossible to synthesize risk of bias or quality concerns of the evidence across reviews despite extracting ROB assessments at the study level as reported by authors of these reviews. For readers, we provide an example of this data in S2 Table for the first 10 randomized controlled trials in alphabetical order to highlight the variability in tools used and reporting.

**Table 1. Overview of review characteristics.**

| Review (author, year) Method of data synthesis | Date of last search (mm/yyyy) | Population (Disease and inclusion criteria) | Tested infusion | Comparator infusion | Antimicrobial(s) studied | Included studies | Mortality | Clinical cure | Treatment failure | Microbiologic cure | Length of stay | Adverse events | Cost | Emergence of resistance | PK/PD outcomes | Critical appraisal and risk of bias |
|---|---|---|---|---|---|---|---|---|---|---|---|---|---|---|---|---|
| Rhodes, 2018 Meta-analysis [25] | 04/2017 | Severely ill / Acutely or critically ill hospitalized patients | PI | II | Piperacillin/tazobactam | RCTs; Observational cohort studies; Retrospective studies; Quasi-experimental studies | ✓ | ✓ | - | ✓ | - | - | - | - | - | Critically low quality[a]; Low risk of bias[b] |
| Vardakas, 2018 Meta-analysis [29] | 11/2016 | Adult patients with sepsis | PI | II | Anti-pseudomonal beta-lactam | RCTs | ✓ | ✓ | - | - | - | ✓ | - | ✓ | - | Low quality[a]; Low risk of bias[b] |
| Yu, 2018 Meta-analysis [32] | 10/2018 | Severe infections | PI | II | Meropenem | RCTs; Observational studies | ✓ | ✓ | - | - | ✓ | ✓ | - | - | - | Low quality[a]; Low risk of bias[b] |
| Lee, 2017 Meta-analysis [54] | 01/2017 | Critically ill patients with predominantly respiratory infections | CI | II | Beta-lactams | RCTs | ✓ | ✓ | - | - | - | - | - | - | ✓ | Critically low quality[a]; High risk of bias[b] |
| Lal, 2016 Meta-analysis [50] | 09/2015 | Nosocomial pneumonia | PI | II | Beta-lactams | RCTs; Non-randomized trial; Retrospective studies | ✓ | ✓ | - | ✓ | - | ✓ | - | - | - | Low quality[a]; Low risk of bias[b] |
| Roberts, 2016 Meta-analysis [16] | 11/2015 | Patients with severe sepsis or septic shock | CI | II | Beta-lactams | RCTs | ✓ | ✓ | - | - | - | - | - | - | - | Low quality review[a]; Low risk of bias[b] |
| Yang, 2016 Meta-analysis [30] | 09/2015 | Unspecified population | PI | II | Piperacillin/Tazobactam | RCTs; Retrospective studies; Prospective studies | ✓ | ✓ | - | - | ✓ | ✓ | ✓ | - | ✓ | Low quality[a]; Low risk of bias[b] |
| Burgess, 2015 Qualitative review [22] | 10/2014 | Unspecified population | PI | II | Cefepime | No restrictions on study design | ✓ | ✓ | - | - | ✓ | ✓ | - | - | ✓ | Critically low quality[a]; High risk of bias[b] |
| Yang, 2015 Meta-analysis [31] | 04/2014 | Unspecified population | PI | II | Piperacillin/Tazobactam | RCTs; Retrospective studies; Prospective studies | ✓ | ✓ | - | ✓ | - | ✓ | - | - | - | Low quality[a]; Low risk of bias[b] |
| Lux, 2014 Meta-analysis [51] | 06/2014 | Hospital acquired pneumonia | PI | II | Beta-lactams | No restrictions | ✓ | ✓ | - | - | - | ✓ | - | ✓ | ✓ | Moderate quality[a]; Low risk of bias[b] |

(*Continued*)

**Table 1.** (Continued)

| Review (author, year) Method of data synthesis | Date of last search (mm/yyyy) | Population (Disease and inclusion criteria) | Tested infusion | Comparator infusion | Antimicrobial(s) studied | Included studies | Outcomes of interest | | | | | | | | | Critical appraisal and risk of bias |
| --- | --- | --- | --- | --- | --- | --- | --- | --- | --- | --- | --- | --- | --- | --- | --- | --- |
| | | | | | | | Mortality | Clinical cure | Treatment failure | Microbiologic cure | Length of stay | Adverse events | Cost | Emergence of resistance | PK/PD outcomes | |
| Teo, 2014 Meta-analysis [24] | 10/2012 | Acute infections in hospitalized adult patients | PI | II | Beta-lactams | RCTs; Retrospective studies; Prospective studies | ✓ | ✓ | - | - | - | ✓ | - | - | - | Low quality a; High risk of bias b |
| Yusuf, 2014 Qualitative review [33] | 05/2014 | Critically ill patients | PI | II | Piperacillin/Tazobactam | No restrictions | - | - | - | - | - | ✓ | - | - | - | Critically low quality a; High risk of bias b |
| Chant, 2013 Meta-analysis [47] | 09/2013 | Critically ill patients | PI | II | Time dependent anti-microbials | Randomized and non-randomized trials | ✓ | - | ✓ | - | ✓ | - | - | - | - | Critically low quality a; Low risk of bias b |
| Falagas, 2013 Meta-analysis [36] | 01/2012 | Unspecified population | PI | II | Carbapenems, piperacillin/tazobactam | Excluded case reports and case series with <10 patients | ✓ | ✓ | - | - | ✓ | ✓ | - | ✓ | - | Critically low quality a; High risk of bias b |
| Hassan, 2013 Meta-analysis [48] | Not reported | Unspecified population | PI | II | Beta-lactams | RCTs | - | ✓ | - | - | - | - | - | - | - | Critically low quality a; High risk of bias b |
| Korbila, 2013 Meta-analysis [49] | 11/2012 | Unspecified population | PI | II | Cephalosporins (3rd, 4th and 5th generation) | Randomized and non-randomized trials | ✓ | ✓ | - | - | - | ✓ | - | ✓ | - | Critically low quality a; High risk of bias b |
| Garcia, 2012 Qualitative review [53] | Not Reported | Patients with acute infections susceptible to piperacillin/tazobactam | PI | II | Piperacillin/tazobactam | Randomized and non-randomized trials | ✓ | ✓ | - | - | - | ✓ | ✓ | - | ✓ | Critically low quality a; High risk of bias b |
| Mah, 2012 Qualitative review [28] | 09/2011 | Adults requiring piperacillin/tazobactam | PI | II | Piperacillin/Tazobactam | Prospective randomized trials; Cohort studies; Retrospective studies | ✓ | ✓ | - | - | - | - | - | - | ✓ | Critically low quality a; High risk of bias b |
| Tamma, 2011 Meta-analysis [27] | 02/2011 | Unspecified population | PI | II | Beta-lactams | RCTs | ✓ | ✓ | - | - | - | ✓ | - | - | - | Moderate quality review a; Low risk of bias b |
| Roberts, 2009 Meta-analysis [26] | 11/2007 | Hospitalized adults with acute infection | PI | II | Beta-lactams | RCTs | ✓ | ✓ | - | - | - | - | - | - | - | Critically low quality a; High risk of bias b |
| Roberts, 2007 Qualitative review [52] | 07/2006 | Serious infections | CI | II | Beta-lactams | No restrictions | ✓ | ✓ | - | - | ✓ | ✓ | - | - | ✓ | Critically low quality a; High risk of bias b |

a Quality as assessed with AMSTAR-2 tool,

b Risk of bias as assessed with ROBIS tool

PI—prolonged infusion; II—intermittent infusion; CI—continuous infusion; RCT—randomized controlled trial

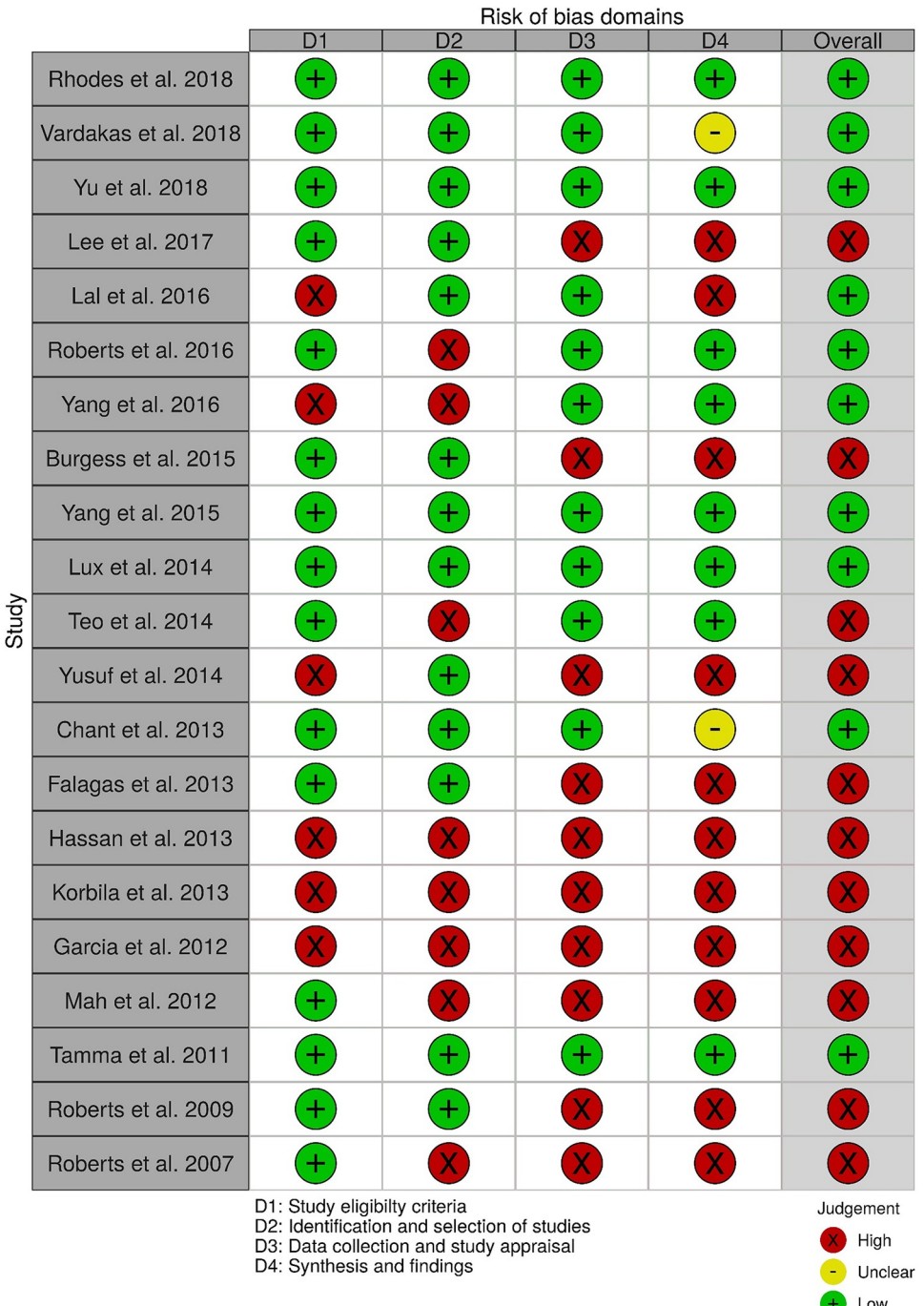

**Fig 2. ROBIS.** Results of the risk of bias evaluation of included systematic reviews using the ROBIS tool (Risk of Bias in Systematic Reviews), categorized per its four domains; D1 -study eligibility criteria, D2 –identification and selection of studies, D3 –data collection and study appraisal, D4 –Synthesis and findings.

## Findings, mortality

Mortality was reported in twenty reviews, with data included from 63 primary studies (S3 Table) [16, 22–33, 36, 47, 49–53]. Overlap of included primary studies across the set of twenty

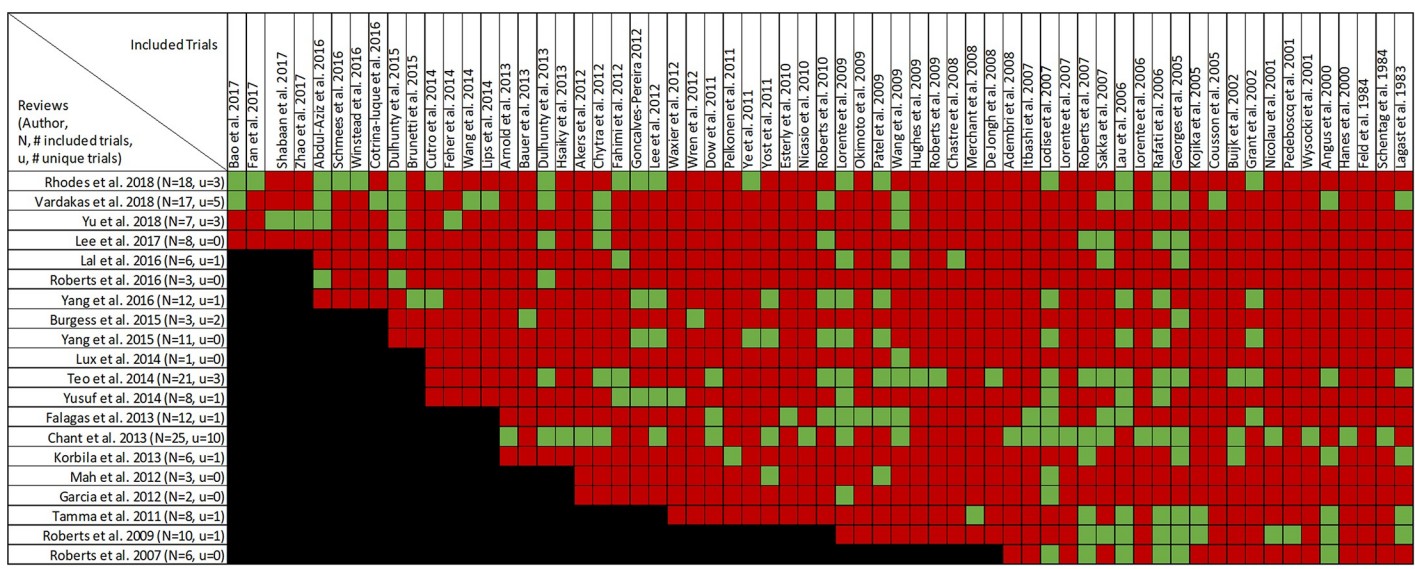

**Fig 3. Citation matrix for reviews reporting mortality of prolonged infusions versus intermittent infusions of beta-lactams.** Green—primary studies included in systematic review, Red—primary study not included in systematic review, Black—primary studies published after systematic review and therefore ineligible for possible inclusion.

reviews was high, with a CCA of 10.4% (Fig 3). Fourteen reviews performed meta-analyses of mortality [16, 23–27, 30–32, 36, 47, 49, 50, 52] while six reviews performed narrative syntheses only [22, 28, 33, 51–53].

Nine of 14 reviews [16, 24, 25, 29–32, 36, 47] show lower mortality with prolonged infusions; the five remaining reviews [23, 26, 27, 49, 50] conveyed uncertainty for effects on mortality, as the confidence intervals included the possibility of no difference and/or favoring control (Fig 4). The magnitude of effect (odds ratios) in reviews describing a mortality

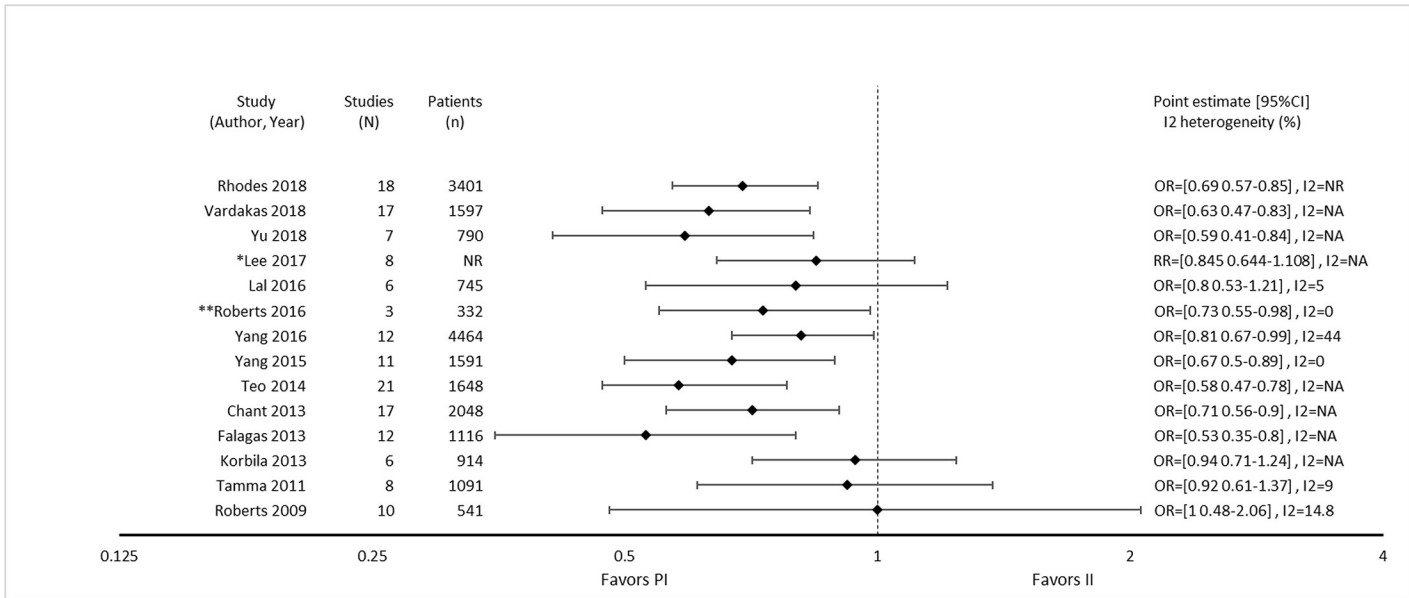

**Fig 4. Effect of prolonged versus intermittent infusions of beta-lactam antimicrobials on mortality.** *Lee 2017 –Did not report event rates. RR was therefore not converted to OR. **Roberts 2016 –individual patient data analysis.

benefit ranged from 0.53 (95% CI: 0.35–0.80) to 0.81(95% CI: 0.67–0.99). Seven of the 9 reviews combined randomized and non-randomized mortality data for this analysis [24, 25, 30–32, 36, 47]. The 9 reviews identifying a mortality benefit were highly variable with respect to scope, quality and risk of bias. Populations of interest included adults who were hospitalized, critically ill, and being treated for severe sepsis, septic shock, and pneumonias where specified. Reviews suggesting a mortality reduction evaluated prolonged infusions of any beta-lactam, classes of beta-lactams such as carbapenems or anti-pseudomonal beta-lactams or specific beta-lactam antibiotics such as piperacillin/tazobactam and meropenem. Only two reviews specifically evaluated cephalosporins and neither identify a mortality benefit [22, 49]. Quality of conduct (per AMSTAR-2) was low or critically low for all 9 reviews identifying a mortality benefit while risk of bias at the review level (per ROBIS) was high for 2 reviews [24, 36] (Fig 2). Eight of the 9 reviews identifying a mortality benefit provide an assessment of risk of bias at the study level with variable findings [16, 24, 25, 29–32, 47]. Pairwise overlap assessment identified the reviews by Yu et al. and Roberts et al. as having slight overlap (<5%) with other reviews [16, 32] (Fig 5). This is likely explained in part by the narrow focus of these reviews where Yu et al. focused only on meropenem while Roberts et al. focused only on patients with severe sepsis and septic shock from three trials in an individual patient data meta-analysis [16, 32].

Of the five reviews failing to describe a mortality benefit, only one combined randomized and non-randomized mortality data (S3 Table) [50]. Amongst these 5 reviews, the populations were similarly diverse including critically ill adults, hospitalized patients with nosocomial pneumonia receiving prolonged or only continuous infusions of beta-lactams or cephalosporin [23, 24, 26, 49, 50]. Only reviews by Lee et al. and Lal et al. were published after 2015 [23, 50]. Only the review by Lee et al. focused on continuous infusions while all others included all prolonged infusions [23]. Quality of conduct (per AMSTAR 2) was low or critically low for all reviews except for the review by Tamma et al. which was scored as moderate [27]. Risk of bias (per ROBIS) was high for 3 of the 5 reviews failing to show a mortality benefit with prolonged infusions of beta lactams (S3 Table) [23, 26, 49]. Only 3 reviews provide risk of bias assessments at the study level (S3 Table) [23, 27, 50]. Pairwise overlap assessment between the 5 reviews that did not identify a mortality benefit show at least medium overlap (≥5%) between all reviews (Fig 4). Pairwise overlap assessment between the 9 reviews that identify a mortality benefit and the 5 reviews that did not identified several combinations with slight (<5%) overlap. The review by Korbila et al. (did not identify a mortality benefit) had slight or no overlap with 6/9 reviews that did identify a mortality benefit [49]. This review focused on 3rd, 4th and 5th generation cephalosporins, only included 6 trials in their mortality evaluation and did not include trials published after November of 2012. Reviews by Yu et al. and Roberts et al., both of whom identified a mortality benefit also had slight or no overlap with three or more negative reviews likely due to issues of scope (Yu et al. included trials of meropenem only while Roberts et al. included trials of patients with severe sepsis or septic shock) [16, 32].

Four reviews assessed the effect of study design on mortality [24, 30, 47, 50]. Three reviews found a significant mortality benefit of prolonged infusions compared to intermittent infusions in when all study designs were combined [24, 30, 47]. However, when randomized and non-randomized trials were pooled separately the mortality benefit was only identified in non-randomized trials. The remaining review failed to identify a mortality benefit when study designs were combined, nor when separated into randomized and non-randomized studies [50]. All authors conclude that mortality data from non-randomized trials may be introducing bias and that only randomized controlled trials should inform this question. It is worth re-mentioning that of the 9 reviews that identified a mortality benefit in this overview seven combined randomized and non-randomized data when pooling.

| | Roberts et al. 2007 | Rhodes et al. 2018 | Vardakas et al. 2018 | Yu et al. 2018 | Lee et al. 2017 | Lal et al. 2016 | Roberts et al. 2016 | Yang et al. 2016 | Burgess et al. 2015 | Yang et al. 2015 | Lux et al. 2014 | Teo et al. 2014 | Yusuf et al. 2014 | Falagas et al. 2013 | Chant et al. 2013 | Korbila et al. 2013 | Mah et al. 2012 | Garcia et al. 2012 | Tamma et al. 2011 |
|---|---|---|---|---|---|---|---|---|---|---|---|---|---|---|---|---|---|---|---|
| Rhodes et al. 2018 | 43% | | | | | | | | | | | | | | | | | | |
| Vardakas et al. 2018 | 44% | 21% | | | | | | | | | | | | | | | | | |
| Yu et al. 2018 | 0% | 9% | 20% | | | | | | | | | | | | | | | | |
| Lee et al. 2017 | 43% | 13% | 35% | 15% | | | | | | | | | | | | | | | |
| Lal et al. 2016 | 14% | 10% | 16% | 10% | 17% | | | | | | | | | | | | | | |
| Roberts et al. 2016 | 0% | 19% | 19% | 33% | 22% | 0% | | | | | | | | | | | | | |
| Yang et al. 2016 | 43% | 47% | 12% | 0% | 11% | 6% | 0% | | | | | | | | | | | | |
| Burgess et al. 2015 | 17% | 0% | 6% | 0% | 10% | 13% | 0% | 0% | | | | | | | | | | | |
| Yang et al. 2015 | 43% | 60% | 14% | 0% | 12% | 6% | 0% | 77% | 0% | | | | | | | | | | |
| Lux et al. 2014 | 0% | 0% | 8% | 33% | 0% | 17% | 0% | 0% | 0% | 0% | | | | | | | | | |
| Teo et al. 2014 | 60% | 32% | 42% | 9% | 33% | 23% | 5% | 28% | 4% | 28% | 5% | | | | | | | | |
| Yusuf et al. 2014 | 50% | 54% | 11% | 0% | 7% | 17% | 0% | 46% | 0% | 46% | 0% | 21% | | | | | | | |
| Falagas et al. 2013 | 22% | 28% | 21% | 8% | 12% | 20% | 0% | 38% | 0% | 35% | 8% | 38% | 18% | | | | | | |
| Chant et al. 2013 | 25% | 16% | 20% | 8% | 23% | 15% | 4% | 17% | 4% | 16% | 4% | 31% | 14% | 19% | | | | | |
| Korbila et al. 2013 | 33% | 0% | 21% | 0% | 18% | 9% | 0% | 0% | 14% | 0% | 0% | 23% | 0% | 0% | 11% | | | | |
| Mah et al. 2012 | 17% | 18% | 0% | 0% | 0% | 0% | 0% | 30% | 0% | 27% | 0% | 10% | 10% | 67% | 9% | 0% | | | |
| Garcia et al. 2012 | 17% | 20% | 0% | 0% | 0% | 14% | 0% | 20% | 0% | 18% | 0% | 10% | 25% | 17% | 9% | 0% | 25% | | |
| Tamma et al. 2011 | 63% | 15% | 42% | 0% | 30% | 8% | 0% | 14% | 11% | 13% | 0% | 30% | 18% | 5% | 13% | 40% | 0% | 0% | |
| Roberts et al. 2009 | 45% | 14% | 50% | 0% | 40% | 15% | 0% | 14% | 10% | 14% | 0% | 37% | 17% | 12% | 24% | 36% | 0% | 0% | 64% |

**Fig 5. Pairwise CCA for reviews reporting mortality of prolonged vs. intermittent infusions of beta-lactams.** Colors indicate degree of overlap, as calculated with CCA, for visual clarity. White = ≤5%, green 5.1–9.9%, yellow 10–14.9%, red ≥15%.

## Findings, clinical cure and treatment failure

Clinical cure and treatment failure was reported in all twenty-one reviews, with data from forty-seven primary studies (S4 Table, Figs 6 and 7) [16, 22–33, 36, 47–53]. Overlap of included primary studies was high (CCA of 13.2%). Fifteen reviews performed quantitative analyses of clinical cure [16, 23–27, 29–32, 36, 48–50] including one review [47] that reported the inverse as treatment failure, while 6 reviews [22, 28, 33, 51–53] report only qualitative syntheses. Nine of 15 reviews combined data from randomized and non-randomized trials [24, 25, 30–32, 36, 47, 49, 50].

Nine of 15 reviews [16, 23–25, 31, 32, 36, 47, 50] show higher clinical cure rates with prolonged versus intermittent infusions; the 6 remaining reviews [26, 27, 29, 30, 48, 49] convey uncertainty (Fig 7). Reviews describing an improvement in clinical cure evaluated prolonged infusions of any beta-lactam and specific agents including piperacillin/tazobactam and meropenem. The two reviews specifically targeting cephalosporins did not show improved clinical

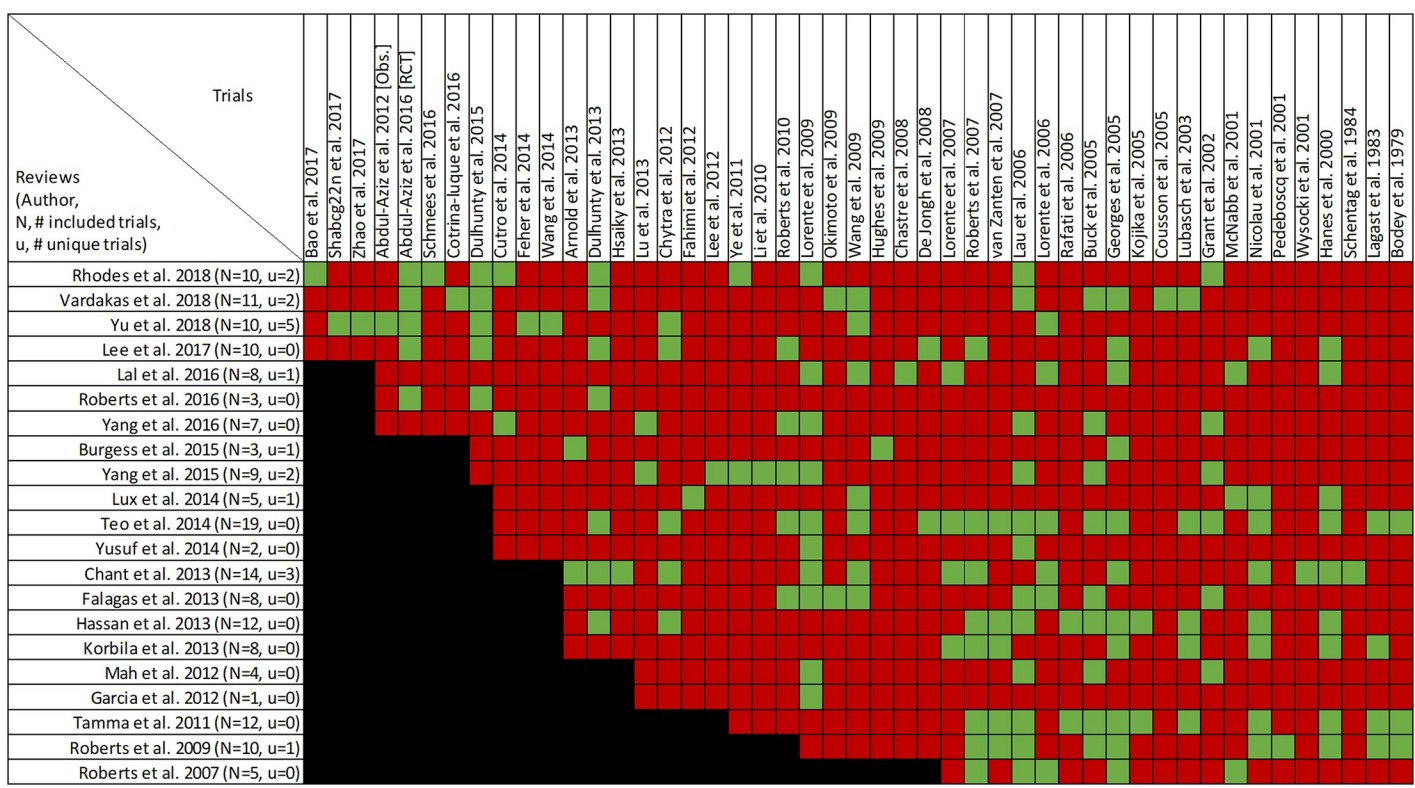

**Fig 6. Citation matrix for reviews reporting clinical cure of prolonged infusions versus intermittent infusions of beta-lactams.** Green—primary studies included in systematic review, Red—primary study not included in systematic review, Black—primary studies published after systematic review and therefore ineligible for possible inclusion.

cure rates with prolonged infusions compared to intermittent infusions, one of which did not perform quantitative synthesis [22, 49]. Among the 9 reviews showing benefit, odds ratio ranged from 1.02 (95% CI: 0.47–2.26) to 2.45 (95% CI: 1.12–20.86) in 8, and relative risk was 1.18 (95% CI: 1.0065–1.3) in 1 review (Fig 7). Six of 9 reviews that identified a benefit with respect to clinical cure were in critically ill populations [23, 25, 47] or patients with severe infections [16, 32, 50], while none of the 6 remaining reviews identify their populations of interest as critically ill or having severe infections. Six reviews conducted a subgroup analysis on mortality risk [16, 23–25, 48, 50]. The five reviews that showed higher rates of clinical cure with prolonged infusions in patients at higher risk of mortality, also showed benefit in their unstratified analyses [16, 23–25, 49]. Three of these reviews also found that patients at lower risk of mortality did have improvements of clinical cure with prolonged infusions [16, 23, 25].

Quality of conduct (per AMSTAR-2) was low or critically low for all 9 reviews identifying a clinical cure benefit, while risk of bias at the review level (per ROBIS) was high for 3 reviews [23, 24, 36] (Fig 2). Sixteen of 21 reviews provide study-level assessments of risk of bias using a variety of tools. Pairwise overlap assessment identified that the individual patient data meta-analysis of 3 trials by Roberts et al. [16] had slight (<5%) overlap with other reviews and that the reviews by Yu et al. [32] and Yang et al. [31] had no overlap as the former focused on meropenem while the latter focused on piperacillin/tazobactam (Fig 8).

Of the 6 reviews failing to identify a clinical cure benefit two combined randomized and non-randomized data [31, 49] (S4 Table). Four of these 6 reviews were published in 2013 or earlier [26, 27, 48, 49]. Quality of reporting (per AMSTAR 2) was low or critically low for all

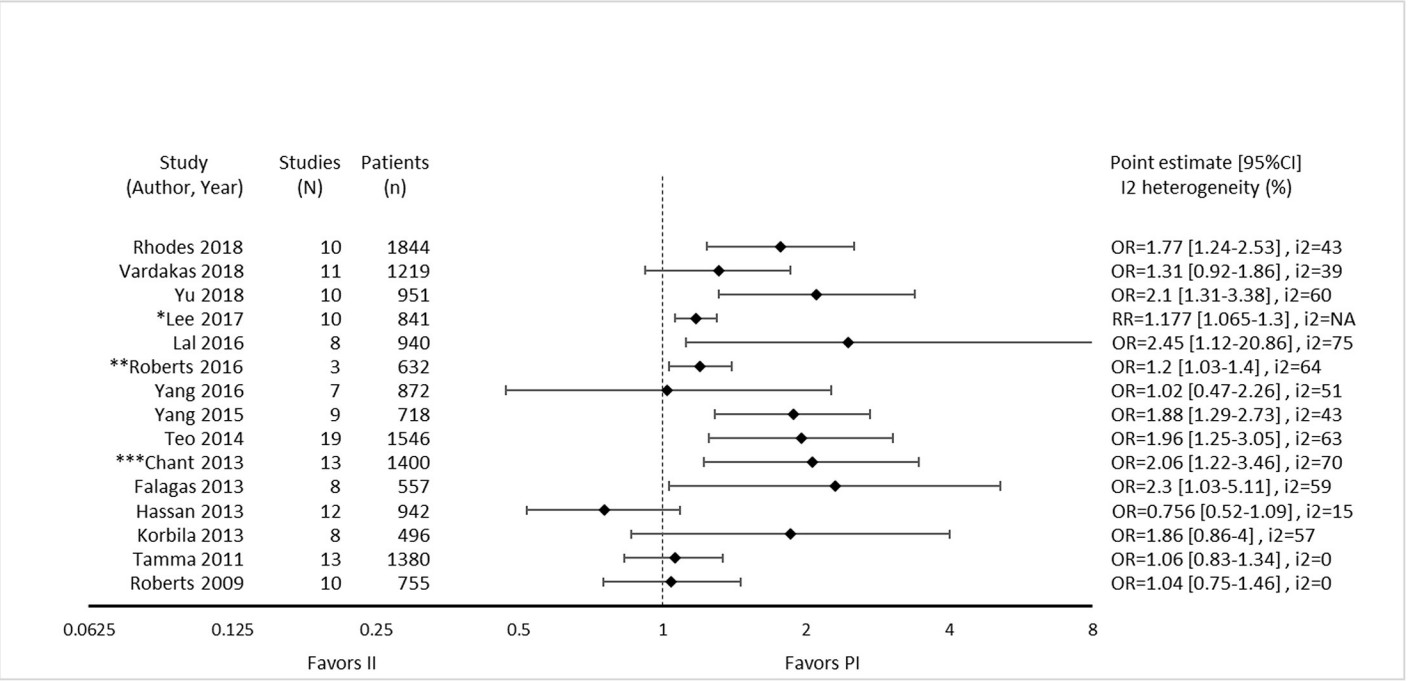

**Fig 7. Effect of prolonged versus intermittent infusions of beta-lactam antimicrobials on clinical cure.** *Lee 2017 –Did not report event rates. RR was therefore not converted to OR. **Roberts 2016 –individual patient data meta-analysis. ***Chant 2013 –reported treatment failure. RR = 0.60 [0.40–0.87], I² = 69%. Results were inverted to represent clinical cure or success.

reviews except for the review by Tamma et al. which was scored as moderate [27]. Risk of bias (per ROBIS) was high for 3 of the 6 reviews failing to show a clinical cure benefit with prolonged infusions of beta lactams [26, 48, 49]. Only 2 reviews did not provide risk of bias assessments at the study level [26, 49]. Pairwise overlap assessment between the 6 reviews that did not identify a clinical cure benefit identified that the reviews of Yang et al. [30] and Korbila et al. [49] had no overlap as the former focused on piperacillin-tazobactam while the latter focused on cephalosporins (Fig 8). Pairwise overlap assessment between the 9 reviews identifying a clinical cure benefit and the 6 that did not, identified several combinations of slight (<5%) to no overlap that can all be explained by differences in scope (i.e., narrow drug focus [32, 49], individual patient data meta-analysis [16]).

Four reviews evaluated the effect of study design on clinical cure [24, 47, 49, 50]. Whereas 3 of 4 reviews showed clinical cure improvements, sub-group analyses of RCTs showed benefit of prolonged infusion in only one review. This review also found no benefit when exclusively assessing cohort studies [47]. Opposing this, subgroup analyses by Teo et al. showed significant benefits in the non-RCT subgroup and no benefit in the RCT subgroup, despite showing improved clinical cure globally [24]. Importantly six reviews did not perform subgroup analyses base on study design as these exclusively included RCTs [16, 23, 26, 27, 29, 48]. Four of these reviews exclusively assessing RCTs did not find significant improvements in clinical cure with prolonged infusions of beta-lactams [26, 27, 29, 48].

## Findings, microbiologic cure

Microbiologic cure was reported in four reviews, with data included from eleven primary studies (S5 Table, Fig 9) [25, 31, 32, 50]. Overlap of included primary studies was moderate, with a CCA of 9.1%. Quantitative analysis of pooled microbiologic cure was performed in all four

| | Roberts et al. 2007 | Rhodes et al. 2018 | Vardakas et al. 2018 | Yu et al. 2018 | Lee et al. 2017 | Lal et al. 2016 | Roberts et al. 2016 | Yang et al. 2016 | Burgess et al. 2015 | Yang et al. 2015 | Lux et al. 2014 | Teo et al. 2014 | Yusuf et al. 2014 | Chant et al. 2013 | Falagas et al. 2013 | Hassan et al. 2013 | Korbila et al. 2013 | Mah et al. 2012 | Garcia et al. 2012 | Tamma et al. 2011 |
|---|---|---|---|---|---|---|---|---|---|---|---|---|---|---|---|---|---|---|---|---|
| Rhodes et al. 2018 | 17% | | | | | | | | | | | | | | | | | | | |
| Vardakas et al. 2018 | 25% | 24% | | | | | | | | | | | | | | | | | | |
| Yu et al. 2018 | 20% | 11% | 17% | | | | | | | | | | | | | | | | | |
| Lee et al. 2017 | 29% | 18% | 24% | 18% | | | | | | | | | | | | | | | | |
| Lal et al. 2016 | 43% | 6% | 12% | 14% | 13% | | | | | | | | | | | | | | | |
| Roberts et al. 2016 | 0% | 33% | 27% | 22% | 63% | 0% | | | | | | | | | | | | | | |
| Yang et al. 2016 | 14% | 33% | 13% | 0% | 42% | 7% | 0% | | | | | | | | | | | | | |
| Burgess et al. 2015 | 20% | 0% | 9% | 0% | 14% | 10% | 0% | 0% | | | | | | | | | | | | |
| Yang et al. 2015 | 14% | 33% | 13% | 0% | 56% | 6% | 0% | 60% | 0% | | | | | | | | | | | |
| Lux et al. 2014 | 14% | 0% | 8% | 11% | 29% | 30% | 0% | 0% | 0% | 0% | | | | | | | | | | |
| Teo et al. 2014 | 29% | 19% | 29% | 14% | 53% | 29% | 5% | 24% | 5% | 22% | 14% | | | | | | | | | |
| Yusuf et al. 2014 | 20% | 33% | 11% | 0% | 0% | 11% | 0% | 29% | 0% | 22% | 0% | 11% | | | | | | | | |
| Chant et al. 2013 | 30% | 12% | 16% | 21% | 38% | 38% | 7% | 5% | 13% | 5% | 19% | 43% | 7% | | | | | | | |
| Falagas et al. 2013 | 29% | 30% | 33% | 22% | 7% | 23% | 0% | 56% | 0% | 42% | 8% | 35% | 25% | 16% | | | | | | |
| Hassan et al. 2013 | 25% | 13% | 33% | 7% | 43% | 11% | 8% | 13% | 7% | 11% | 13% | 48% | 8% | 30% | 11% | | | | | |
| Korbila et al. 2013 | 18% | 0% | 14% | 0% | 33% | 23% | 0% | 0% | 10% | 0% | 18% | 42% | 0% | 29% | 0% | 43% | | | | |
| Mah et al. 2012 | 14% | 60% | 22% | 0% | 83% | 9% | 0% | 67% | 0% | 44% | 0% | 22% | 50% | 7% | 50% | 15% | 0% | | | |
| Garcia et al. 2012 | 0% | 25% | 0% | 0% | 0% | 13% | 0% | 17% | 0% | 11% | 0% | 6% | 50% | 9% | 13% | 0% | 0% | 25% | | |
| Tamma et al. 2011 | 21% | 7% | 27% | 0% | 64% | 11% | 0% | 13% | 8% | 12% | 14% | 53% | 8% | 22% | 11% | 83% | 54% | 14% | 0% | |
| Roberts et al. 2009 | 25% | 8% | 21% | 0% | 88% | 13% | 0% | 17% | 9% | 17% | 17% | 53% | 9% | 25% | 13% | 54% | 50% | 17% | 0% | 69% |

**Fig 8. Pairwise CCA of reviews reporting clinical cure of prolonged versus intermittent infusions of beta-lactams.** Colors indicate degree of overlap, as calculated with CCA, for visual clarity. White = ≤5%, green 5.1–9.9%, yellow 10–14.9%, red ≥15%.

reviews. A definition for microbiologic cure was provided in only one review as "eradication or presumed eradication" as compared to "persistence or presumed persistence" without further explanation [31]. All four reviews combined randomized and non-randomized evidence.

Only the review by Yu et al. demonstrated a statistically significant benefit regarding microbiologic cure with prolonged infusions [32] (Fig 10). Notably, this review focused on meropenem prescribed for acute infections while the reviews by Rhodes et al. and Yang et al. [25, 31] focused only on piperacillin/tazobactam and the review by Lal et al. [50] included all beta lactam antimicrobials but limited infections to nosocomial pneumonia. Despite the overall moderate overlap, pairwise overlap assessment confirms that the review by Yu et al. [32] included four primary studies [55–58] not found in the other three reviews and has no overlap with either of the 3 reviews that did not find a microbiological cure benefit with prolonged infusions when compared with intermittent administration (Fig 11). Predictably only the reviews by Yang et al. and Rhodes et al. had overlap (60%) given that they both focused on piperacillin/tazobactam [25, 31].

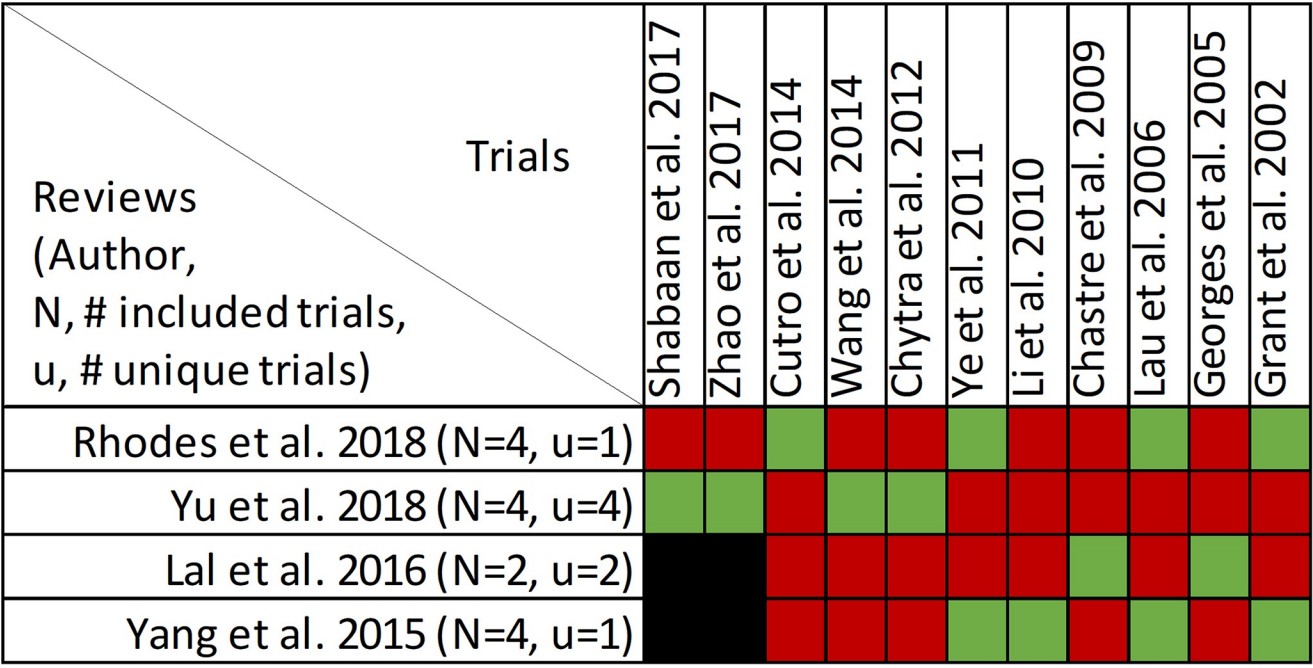

**Fig 9. Citation matrix for reviews reporting microbiologic cure of prolonged infusions.** Green—primary studies included in systematic review, Red— primary study not included in systematic review, Black—primary studies published after systematic review and therefore ineligible for possible inclusion.

Quality of conduct (per AMSTAR-2) was low or critically low for all 4 reviews evaluating microbiologic cure while risk of bias at the review level (per ROBIS) was low for all 4 reviews [25, 31, 32, 50] (Fig 2). Study level risk of bias was assessed using a variety of tools (Cochrane Risk of Bias Tool, Jadad Scale, Newcastle Ottawa Scale) in all 4 reviews overall but none at the outcome level. The review by Lal et al. stated that they used the Jadad and Newcastle Ottawa Scale but the data are not reported [50].

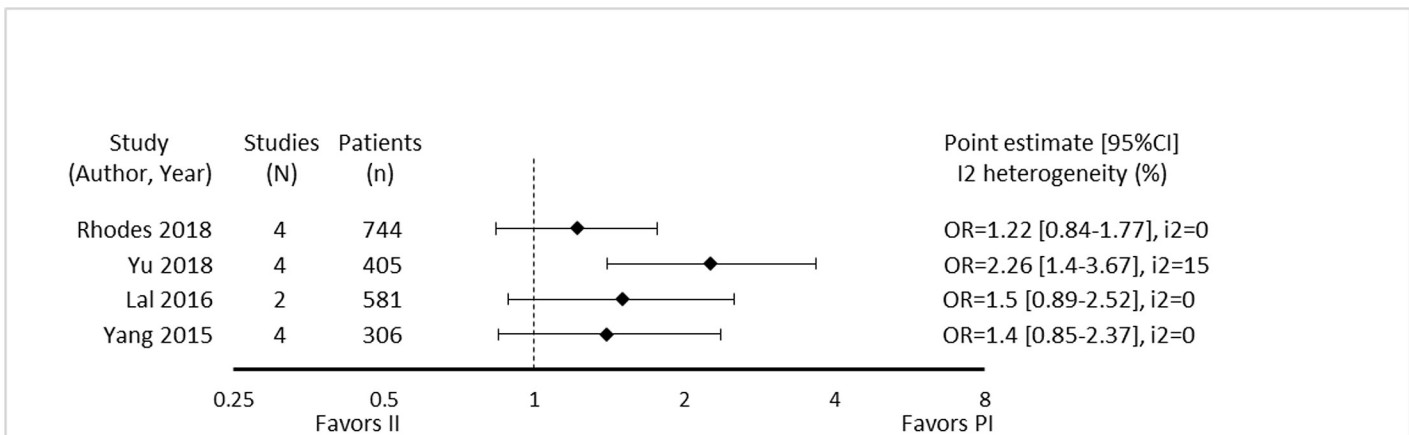

**Fig 10. Effect of prolonged infusions versus intermittent infusions of beta-lactam antimicrobials on microbiologic cure.**

**Fig 11. Pairwise CCA for reviews reporting microbiologic cure of prolonged versus intermittent beta-lactam infusions.** Colors indicate degree of overlap, as calculated with CCA, for visual clarity. White = ≤5%, green 5.1–9.9%, yellow 10–14.9%, red ≥15%.

### Findings, length of stay

Length of stay was reported in eight reviews, with data included from 37 primary studies (S6 Table, Fig 12) [22, 25, 28, 30, 32, 33, 47, 52]. Overlap of included primary studies was moderate with CCA of 7.7%. Two reviews assessed length of stay quantitatively [25, 47] while six reviews provide only qualitative assessments [34, 46, 47, 53, 57, 62].

Neither of the two quantitative analyses showed a significant reduction in hospital or ICU LOS [25, 47]. In the review by Rhodes et al. focusing on piperacillin/tazobactam in critically ill patients, the ratio of means from 12 studies evaluating hospital length of stay in 2,916 patients showed a numerical lower, but insignificant, reduction with prolonged infusions 0.92 (95% CI: 0.83–1.03; $I^2 = 39\%$) [25]. The ratio of means from 10 studies evaluating ICU length of stay in 2593 patients was reported as 1.04 (95% CI: 0.87–1.25; $I^2 = 66.2\%$). Subgroup analysis by risk of mortality (20% threshold) also showed no benefit of prolonged infusions when compared

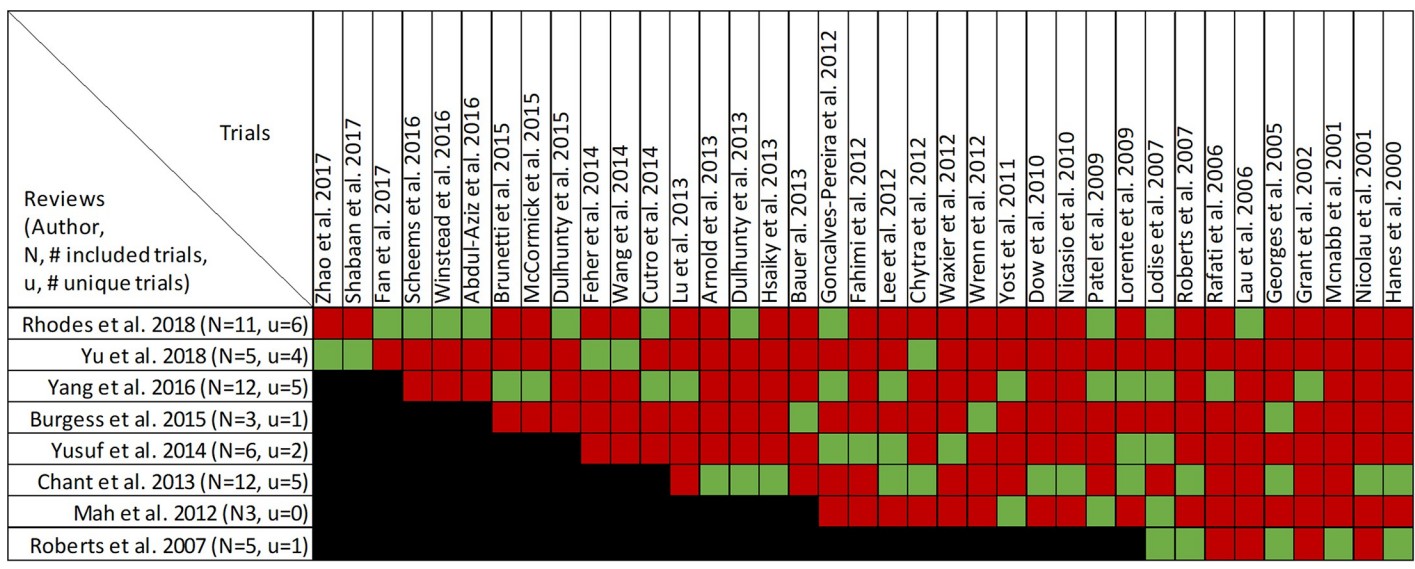

**Fig 12. Citation matrix for reviews reporting length of stay of prolonged infusions versus intermittent infusions of beta-lactams.** Green—primary studies included in systematic review, Red—primary study not included in systematic review, Black—primary studies published after systematic review and therefore ineligible for possible inclusion.

with intermittent infusions. The review by Chant et al. evaluating prolonged infusions of all time dependent antibiotics (one trial each of prolonged infusions of linezolid and vancomycin were included) in critically ill patients report a mean difference from 9 studies enrolling 1417 patients of -1.08 days (95% CI: -4.25 to 2.10; $I^2 = 56\%$) [47]. While statistically non-significant, the trend towards benefit with prolonged infusions appears to be driven by non-randomized trials. The mean difference from 4 randomized trials (n = 315) was 4.90 days (95% CI: -1.83 to 11.64; $I^2 = 8\%$), while the mean difference from 5 non-randomized trials (n = 1102) was -2.38 days (95% CI: -5.04 to 0.28; $I^2 = 29\%$). The mean difference in ICU length of stay between prolonged infusions from 11 trials enrolling 1495 patients in the review by Chant et al. was -1.02 days (95% CI: -2.65 to 0.60; $I^2 = 60\%$) [47]. In this case the sensitivity analysis by study design did identify a statistically significant benefit with prolonged infusions on ICU length of stay whereby the mean difference among 5 randomized trials (n = 444) was -1.5 days (95% CI: -2.81 to -0.19; $I^2 = 0$). The mean difference in ICU length of stay from 6 non-randomized trials (n = 1053) was not statistically significant at -0.86 days (95% CI: -3.60 to 1.88; $I^2 = 73\%$). Pairwise overlap assessment between these two reviews reveal moderate overlap with the review by Rhodes et al. focusing on only piperacillin/tazobactam while the review by Chant (2013) et al included a much broader range of time dependent antibiotics [25, 47] (Fig 13). Furthermore, given the 5 year difference in publication dates 12 trials included by other reviews were published after Chant et al in 2013.

Quality of conduct (per AMSTAR-2) was critically low for both reviews with quantitative evaluations of length of stay [25, 47]. Risk of bias at the review level (per ROBIS) was low for both reviews (Fig 2) [25, 47]. Study level risk of bias was assessed in both reviews but the tool used by Chant et al. is not clear [47]. ROB was assessed by neither review at the outcome level.

### Findings, adverse events

Adverse events were reported in 14 reviews, with data included from 34 primary studies (S7 Table, Fig 14) [22, 24, 27–29, 31–33, 49–53, 59]. Overlap of included studies was slight, with a

|  | Roberts et al. 2007 | Rhodes et al. 2018 | Yu et al. 2018 | Yang et al. 2016 | Burgess et al. 2015 | Yusuf et al. 2014 | Chant et al. 2013 |
|---|---|---|---|---|---|---|---|
| Rhodes et al. 2018 | 17% | | | | | | |
| Yu et al. 2018 | 0% | 0% | | | | | |
| Yang et al. 2016 | 14% | 22% | 0% | | | | |
| Burgess et al. 2015 | 20% | 0% | 0% | 0% | | | |
| Yusuf et al. 2014 | 50% | 20% | 0% | 33% | 0% | | |
| Chant et al. 2013 | 50% | 6% | 8% | 5% | 7% | 13% | |
| Mah et al. 2012 | 20% | 40% | 0% | 38% | 0% | 13% | 0% |

**Fig 13. Pairwise CCA for reviews reporting length of stay of prolonged versus intermittent infusions of beta-lactams.** Colors indicate degree of overlap, as calculated with CCA, for visual clarity. White = ≤5%, green 5.1–9.9%, yellow 10–14.9%, red ≥15%.

CCA of 4.8% (acknowledging that the review by Vardakas et al. did not identify the trials they included in the evaluation of this outcome) [29].

The review by Vardakas et al. found no difference after meta-analysis of adverse events rates from seven unspecified RCTs enrolling 980 patients (RR 0.88; 95%CI: 0.71–1.09:

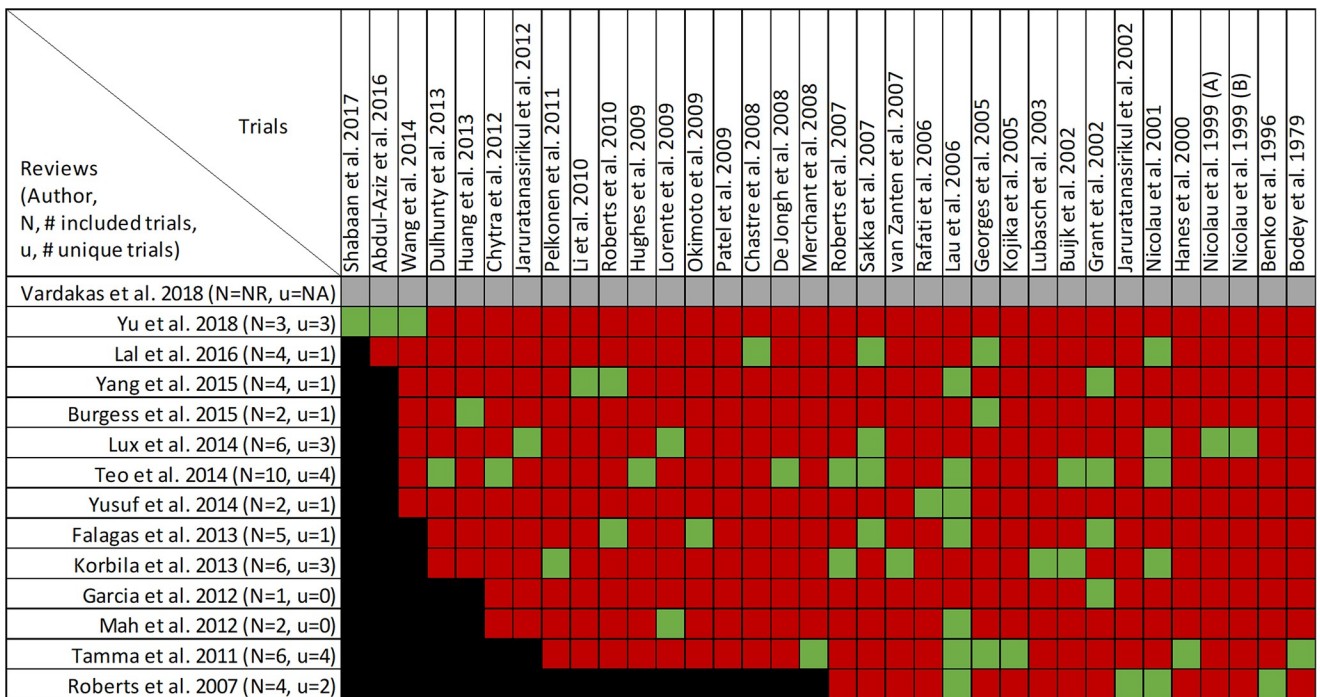

**Fig 14. Citation matrix for review reporting adverse events of prolonged infusions versus intermittent infusions of beta-lactams.** Green—primary studies included in systematic review, Red—primary study not included in systematic review, Black—primary studies published after systematic review and therefore ineligible for possible inclusion, Grey—included primary studies not reported for the specified outcome.

$I^2 = 0\%$) [29]. Thirteen reviews qualitatively described findings from primary studies. All reviews described studies that did not find a difference in rate of adverse events [22, 24, 27, 28, 31–33, 49–53, 59]. Of these, three reviews [21, 28, 52] described the findings of one study [60] that attributed serious adverse to 6 patients with complicated intra-abdominal infections randomized to receive a continuous infusion of piperacillin/tazobactam including *Clostridium difficile* colitis, renal failure, confusion, tachycardia, and a tonic/clonic seizure. This study did not find significant differences in the rate of serious adverse events between continuous and intermittent infusion groups and none of the adverse events were associated with death. One review [32] described a significant increase in acute kidney injury with intermittent versus prolonged infusion of meropenem in neonates in one trial [56]. Tamma et al. noted that almost all studies did not define adverse events *a priori* [27]. The extreme variability in rate of adverse events was also noted in one review, where rate of adverse events of included studies ranged from 0% to 89% [52]. Pairwise overlap assessment between the review by Vardakas et al. and the other 13 reviews with narrative assessments of adverse events was inconclusive as Vardakas et al. did not identify the individual trials informing this outcome (Fig 15) [29].

Quality of conduct (per AMSTAR-2) for the Vardakas et al. review was low as was their risk of bias (per ROBIS) [29]. The quality of reporting and risk of bias for the remaining reviews varied widely. Study level risk of bias was not performed in 3 of 14 reviews [22, 36, 49] and no review conducted a risk of bias assessment at the outcome level.

| | Roberts et al. 2007 | Vardakas et al. 2018 | Yu et al. 2018 | Lal et al. 2016 | Yang et al. 2015 | Burgess et al. 2015 | Lux et al. 2014 | Teo et al. 2014 | Yusuf et al. 2014 | Falagas et al. 2013 | Korbila et al. 2013 | Garcia et al. 2012 | Mah et al. 2012 |
|---|---|---|---|---|---|---|---|---|---|---|---|---|---|
| Vardakas et al. 2018 | n/a | | | | | | | | | | | | |
| Yu et al. 2018 | 0% | n/a | | | | | | | | | | | |
| Lal et al. 2016 | 17% | n/a | 0% | | | | | | | | | | |
| Yang et al. 2015 | 20% | n/a | 0% | 0% | | | | | | | | | |
| Burgess et al. 2015 | 0% | n/a | 0% | 20% | 0% | | | | | | | | |
| Lux et al. 2014 | 14% | n/a | 0% | 25% | 0% | 0% | | | | | | | |
| Teo et al. 2014 | 25% | n/a | 0% | 17% | 17% | 0% | 14% | | | | | | |
| Yusuf et al. 2014 | 20% | n/a | 0% | 0% | 20% | 0% | 0% | 9% | | | | | |
| Falagas et al. 2013 | 17% | n/a | 0% | 13% | 50% | 0% | 10% | 25% | 17% | | | | |
| Korbila et al. 2013 | 13% | n/a | 0% | 11% | 0% | 0% | 9% | 23% | 0% | 0% | | | |
| Garcia et al. 2012 | 0% | n/a | 0% | 0% | 25% | 0% | 0% | 11% | 0% | 20% | 0% | | |
| Mah et al. 2012 | 25% | n/a | 0% | 0% | 20% | 0% | 14% | 10% | 33% | 17% | 0% | 0% | |
| Tamma et al. 2011 | 13% | n/a | 0% | 11% | 11% | 17% | 0% | 8% | 14% | 10% | 0% | 0% | 14% |

**Fig 15. Pairwise CCA for reviews reporting adverse events or prolonged versus intermittent infusions of beta-lactams.** Colors indicate degree of overlap, as calculated with CCA, for visual clarity. White = ≤5%, green 5.1–9.9%, yellow 10–14.9%, red ≥15%. N/A indicated where primary studies included were not reported by systematic review.

### Findings, cost

Costs of prolonged versus intermittent infusions of beta-lactams were reported in five reviews, with data included from 9 primary studies (S8 Table, Fig 16) [28, 30, 31, 52, 53]. Overlap of included primary studies was moderate, with a CCA of 8.3%. One review by Yang et al. that focused on piperacillin/tazobactam in an unspecified population performed a meta-analysis of healthcare cost data from three studies (n = 2298) and described mean difference of -0.38 (95% CI: -0.70 to -0.07; $I^2$ = 61%) in favor of prolonged infusions when compared with traditional dosing [27]. Further sensitivity analysis by study design reveal that this cost benefit in favor of prolonged infusions was evident in one randomized trial (n = 50; mean difference -0.94; 95% CI: -1.52 to -0.35) as well as in two non-randomized studies (n = 2248; mean difference -0.25; 95% CI: -0.34 to -0.16; $I^2$ = 0%).

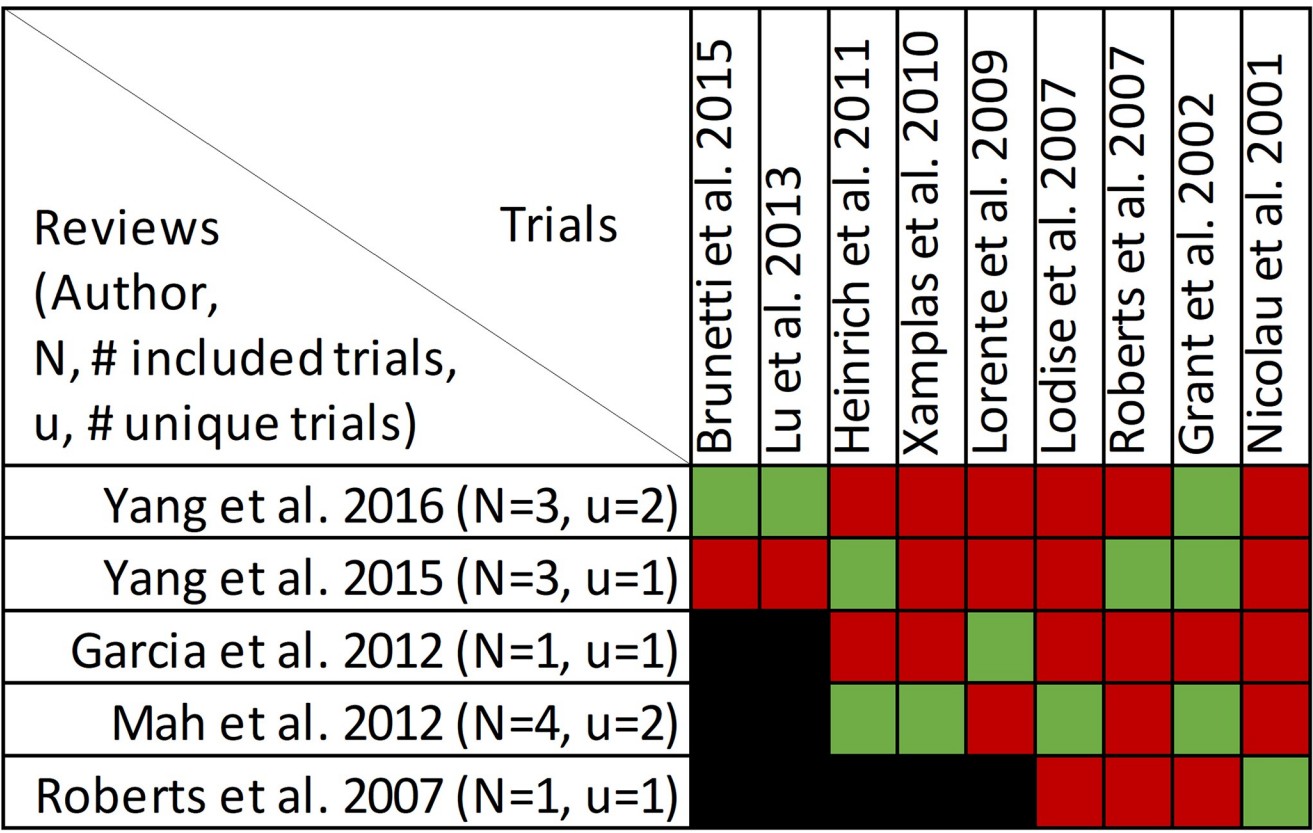

**Fig 16. Citation matrix for reviews reporting cost of prolonged versus intermittent infusions of beta-lactams.** Green—primary studies included in systematic review, Red—primary study not included in systematic review, Black—primary studies published after systematic review and therefore ineligible for possible inclusion.

In other reviews with descriptive reporting Roberts et al. report on one primary study that showed significant cost reduction in aggregate costs related to drug procurement, preparation, administration, adverse events and therapeutic failure [52]. Similarly, Garcia et al. report on one other primary study that showed reduction in aggregate costs related to drug preparation, administration, adverse events and therapeutic failure, but not drug procurement costs [53] related to prolonged infusions. Two reviews only discuss cost implications of prolonged infusions in their discussion sections. Both reviews highlighted positive findings with regards to cost, however no conclusions are provided by the authors and the presented findings do not appear to be the product of a systematic review of cost related outcomes of included studies [28, 31]. Pairwise overlap assessment suggests that the review by Yang (2016) has very high overlap with their prior review (Yang (2015) et al) and Mah et al. (Fig 17) [28, 30, 31].

Quality of conduct (per AMSTAR-2) for the Yang et al. review was low as was their risk of bias (per ROBIS) [30]. The quality of reporting and risk of bias for the remaining reviews varied widely. Study level risk of bias was performed by all reviews but not at the outcome level.

### Findings, emergence of antimicrobial resistance

Emergence of resistance with prolonged infusions of beta-lactams was evaluated in four reviews, with data included from nine primary studies (S9 Table, Fig 18) [29, 36, 49, 51]. Overlap of

|  | Roberts et al. 2007 | Yang et al. 2016 | Yang et al. 2015 | Garcia et al. 2012 |
|---|---|---|---|---|
| Yang et al. 2016 | 0% | | | |
| Yang et al. 2015 | 0% | 20% | | |
| Garcia et al. 2012 | 0% | 0% | 0% | |
| Mah et al. 2012 | 0% | 25% | 40% | 0% |

**Fig 17. Pairwise CCA for reviews reporting cost of prolonged versus intermittent infusions of beta-lactams.** Colors indicate degree of overlap, as calculated with CCA, for visual clarity. White = ≤5%, green 5.1–9.9%, yellow 10–14.9%, red ≥15%. N/A indicated where primary studies included were not reported by systematic review.

included studies moderate with a CCA of 7.4% (of note, the review by Vardakas [29] et al. did not identify the individual trials informing this outcome and this is reflected in the CCA calculation). The relationship between antimicrobial resistance was evaluated descriptively in all reviews except for the review by Vardakas et al. which focused on anti-pseudomonal beta-lactam antibiotics in septic patients [29]. In their review, two unidentified randomized trials showed no difference in the development of resistance between prolonged and intermittent infusions (RR 0.60, 95% CI: 0.15 to 2.38) [29]. They also report that two other unidentified randomized trials identified no resistant strains in either treatment group [29]. In the review by Falagas et al. evaluating carbapenems and piperacillin/tazobactam in an unspecified population

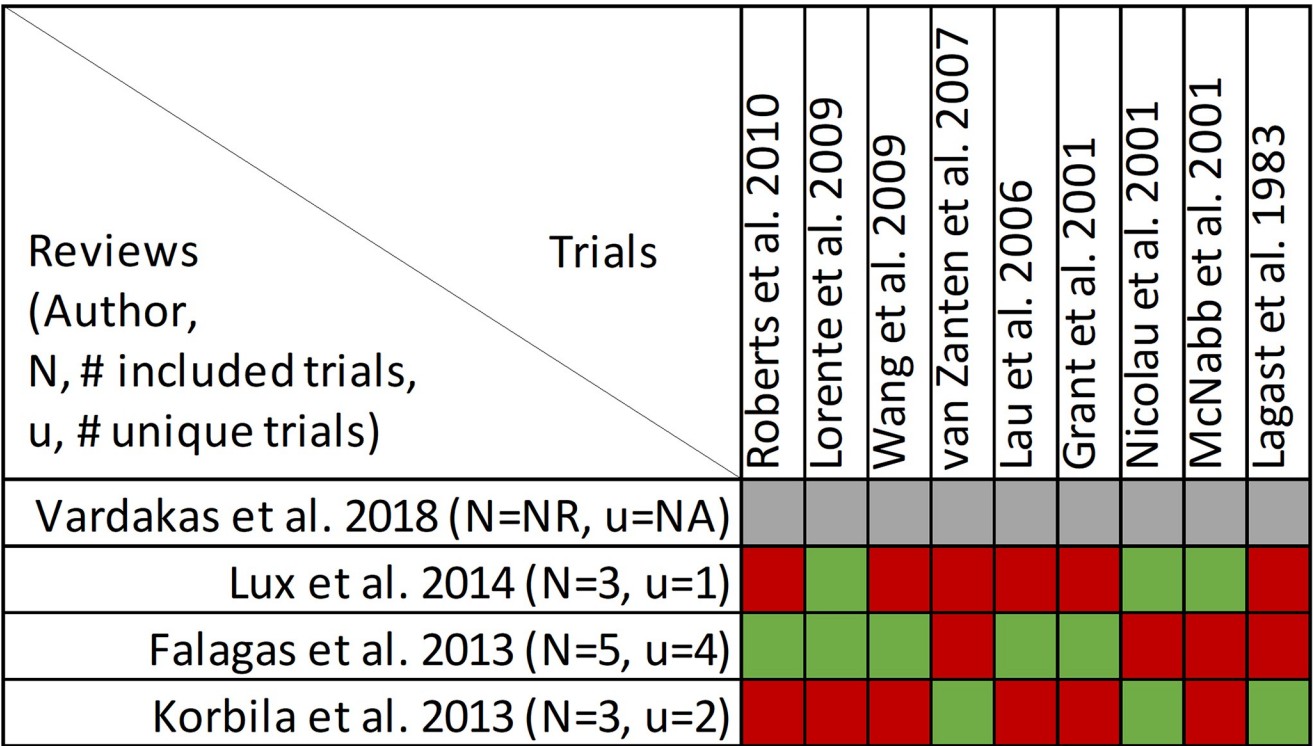

**Fig 18. Citation matrix for reviews reporting emergence of resistance of prolonged versus intermittent infusion of beta-lactams.** Green—primary studies included in systematic review, Red—primary study not included in systematic review, Black—primary studies published after systematic review and therefore ineligible for possible inclusion, Grey—included primary studies not reported for the specified outcome.

five studies were identified, four of which did not identify antimicrobial resistance in any treatment group [36]. They identified that one trial [61] reported two clinical isolates that developed resistance during therapy with continuous infusions of piperacillin/tazobactam but the quality of reporting was insufficient for evaluation. Lux et al. intended to evaluate antibiotic resistance in their review involving all beta lactams used for hospital acquired pneumonia but deemed that one RCT and one retrospective cohort that they identified to have insufficient evidence for analysis [51]. Finally the review by Korbila et al. that evaluated cephalosporine infusions in an unspecified population identified 3 trials that evaluated antimicrobial resistance, from which only one trial [62] found that 12% of patients in the intermittent infusion arm developed resistant isolates compared with none in the extended or continuous infusion group [49]. Given the variability in scope between these review pairwise assessment of overlap identified that only the review by Lux et al., which evaluated all beta lactams, had identifiable overlap with reviews by Korbila et al. and Falagas et al. (Fig 19) [36, 49, 51].

Quality of conduct (per AMSTAR-2) for the Vardakas et al. review was low as was their risk of bias (per ROBIS) [29]. The quality of reporting and risk of bias for the remaining reviews varied widely. Study level risk of bias was performed by all reviews but not at the outcome level.

### Findings, pharmacokinetic/pharmacodynamic outcomes

PK/PD outcomes were reported in six reviews, with data from 29 primary studies (S10 Table, Fig 20) [22, 23, 28, 51–53]. Overlap of included studies was slight, with mCCA of 4.8%. Of

| | Korbila et al. 2013 | Vardakas et al. 2018 | Lux et al. 2014 |
|---|---|---|---|
| Vardakas et al. 2018 | n/a | | |
| Lux et al. 2014 | 20% | n/a | |
| Falagas et al. 2013 | 0% | n/a | 14% |

**Fig 19. Pairwise CCA for reviews reporting emergence of resistance of prolonged versus intermittent infusion of beta-lactams.** Colors indicate degree of overlap, as calculated with CCA, for visual clarity. White = ≤5%, green 5.1–9.9%, yellow 10–14.9%, red ≥15%. N/A indicated where primary studies included were not reported by systematic review.

these 6 reviews only 3 report the outcomes of interest: $f$T>MIC and the probability of PD target attainment [22, 23, 28]. None performed meta-analysis of this outcome.

The review by Lee et al. evaluating beta-lactam antibiotics in critically ill patients with respiratory infections identified 11 trials evaluating $f$T>MIC [23]. In these 11 trials the $f$T>MIC for the prolonged infusion treatment arms was 97–100% while in the intermittent bolus arms the $f$T>MIC ranged from 46–100% using MICs of susceptible strains. The authors report that statistically significant differences were only found in 3 of 11 trials all in favor of continuous

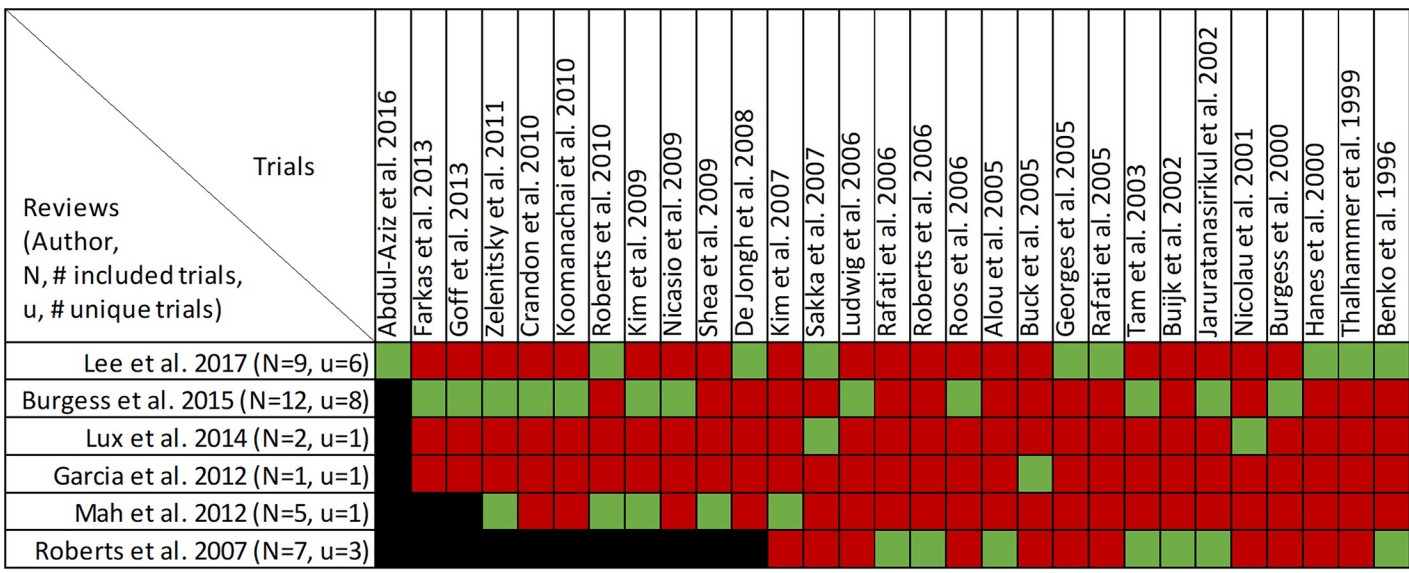

**Fig 20. Citation matrix for reviews reporting PK/PD outcomes of prolonged versus intermittent infusions of beta-lactams.** Green—primary studies included in systematic review, Red—primary study not included in systematic review, Black—primary studies published after systematic review and therefore ineligible for possible inclusion.

infusions [23]. The review by Burgess et al. only included trials of cefepime and identified 3 trials in critically ill patients reporting $f$T>MIC and the probability of target (>50% $f$T>MIC) attainment using Monte Carlo simulation [22]. In these trials the $f$T>MIC ranged from 50 to 65% and the simulation studies suggest that 4g per day administered as a continuous or prolonged infusion would be required for >90% target attainment [22]. The review by Mah et al. focusing on piperacillin/tazobactam identified 4 trials that compare the probability of target attainment (>50% $f$T>MIC) [28]. Amongst these trials, prolonged or continuous infusion of at least 13.5g per day were required to achieve >90% probability of target attainment in a variety of hospitalized patients. No dosing strategy involving traditional infusions of piperacillin/tazobactam achieved this target of >90% probability [28]. Pairwise overlap assessment between these 3 reviews reveal high overlap between reviews by Burgess et al. and Mah et al. and moderate overlap between reviews by Lee et al. and Mah et al. (Fig 21) [22, 23, 28]. The latter is expected due to similarities in scope despite the 5 years between publications. The overlap between Burgess et al. and Mah et al. is unexpected given the focus on different antimicrobials [22, 28]. This evaluation of overlap may be overestimated due to the fact that some of the primary studies included in both reviews provide PK data on multiple antimicrobials.

Quality of reporting (per AMSTAR-2) for all three reviews discussed were critically low. Risk of bias (per ROBIS) was high for reviews by Lee et al. and Mah et al. and low for the review by Burgess et al. [22, 23, 28]. Study level risk of bias was performed in all reviews except Burgess et al. and no review performed an assessment of risk of bias at the outcome level.

## Discussion

Our overview of systematic reviews identified 21 reviews evaluating effects of prolonged infusions of beta-lactams on clinical cure or mortality. Considerable variability was seen across populations, beta-lactams, infusion protocol and outcomes and their definitions. These differences preclude us from broadly generalizing the benefits of prolonged infusions of beta-lactam antimicrobials. Rather, the current body of systematic reviews and primary studies present a

| | Roberts et al. 2007 | Lee et al. 2017 | Burgess et al. 2015 | Lux et al. 2014 | Garcia et al. 2012 |
|---|---|---|---|---|---|
| Lee et al. 2017 | 8% | | | | |
| Burgess et al. 2015 | 20% | 0% | | | |
| Lux et al. 2014 | 0% | 11% | 0% | | |
| Garcia et al. 2012 | 0% | 0% | 0% | 0% | |
| Mah et al. 2012 | 0% | 8% | 15% | 17% | 0% |

**Fig 21. Pairwise CCA for reviews reporting PK/PD outcomes of prolonged versus intermittent infusion of beta-lactams.** Colors indicate degree of overlap, as calculated with CCA, for visual clarity. White = ≤5%, green 5.1–9.9%, yellow 10–14.9%, red ≥15%. N/A indicated where primary studies included were not reported by systematic review.

fragmented picture of the use and impact of prolonged infusions of beta-lactam antimicrobials in different clinical contexts. It does appear that with respect to the outcomes of mortality and clinical cure, the populations where benefits were observed most often were critically ill patients or those with serious infections. Furthermore, the sub-classes of beta-lactams most often studied were penicillins (particularly piperacillin-tazobactam) and carbapenems (particularly meropenem). Most reviews combined extended and continuous infusions under the umbrella of prolonged infusions, and thus there was insufficient evidence to comment on differences between infusion strategies. Although the reviews that reported on PD target attainment identified that prolonged infusions were significantly more likely to result in PD target attainment when compared to traditional dosing, no review compared the probability of PD target attainment between extended and continuous infusions. The outcomes of interest with the most evidence were mortality and clinical cure. Mortality was improved with prolonged infusions of beta lactams in most reviews in a variety of hospitalized patients including critically ill and those with pneumonias. Reviews that focused on cephalosporins were few and did not identify a mortality benefit. However, among reviews that evaluated study design in sensitivity analyses, all authors suggest that meta-analyses of non-randomized studies may introduce bias due to residual confounding and selection bias. It is important to recognize that 7 of 9 reviews that identified a mortality benefit combined randomized and non-randomized data, while 4 of 5 reviews that did not identify a mortality benefit only evaluated randomized controlled trials. Similarly, most (9 of 15) reviews evaluating clinical cure (albeit variably defined) describe a benefit in patients receiving prolonged infusions as compared with intermittent infusions, however bias introduced by including non-randomized studies may have influenced this outcome. With both of these outcomes, the reviews that conducted subgroup analyses by severity of illness suggest that benefit may be greater in more severely ill patients as compared with less sick patients. The other outcomes of interest including microbiologic cure, length of stay, adverse events, cost, emergence of antimicrobial resistance and PK/PD outcomes were less well informed. Microbiologic cure was only improved by prolonged infusions of meropenem in one of four reviews. Two reviews evaluating length of stay quantitatively did not find a difference between prolonged and intermittent infusions. Adverse events were poorly reported in primary studies and thus reviews were also inconsistent in their evaluation of adverse events. The one review that performed a meta-analysis of cost data did associate a healthcare cost savings with prolonged infusions from 3 trials. While 4 reviews addressed antimicrobial resistance, the effect of prolonged infusions of beta lactams on this outcome remains uncertain. Finally, despite only 3 reviews addressing PK/PD outcomes, prolonged infusions appear to maximize $f$T>MIC, and this may lead to a greater chance of PD target attainment. The evidence describing the relationship between target attainment and clinical outcomes was insufficient. While mortality and clinical cure outcomes were well represented in these reviews, we strongly feel that all these outcomes are important and future reviews and trials should address them with equal attention.

It must be recognized that the quality of reviews and risk of bias assessment at the study level and the review level was variable but generally poor. Poor quality conduct of reviews impedes decision making and represents waste in use of research dollars [63, 64]. Assessing the risk of bias of included evidence is an important step of conduct for systematic reviews, for which the evidence needs to be properly contextualized for the reader to understand to what extent they should be placing confidence in those findings. The GRADE framework is an internationally-endorsed best practice approach for understanding the certainty of evidence findings that formally includes the risk of bias assessment of the body of evidence along with other considerations to evaluate the confidence or certainty of the evidence. Undertaking GRADE assessments is integral to systematic review work to 'set-the-stage' for the use of findings in healthcare decision-making. With the exception of Roberts et al. [26] and Tamma et al. [27], it

is conceivable that most reviews could have made use of GRADE methodology to interpret the evidence for readers. This means that, despite reports of effectiveness in some reviews, readers are left without a complete assessment of whether those results have sufficient certainty for applying in practice. Therefore, we suggest that future authors of systematic reviews consult GRADE's published guidance or their online or in-person training offerings to implement use of this methodology. Given poor adherence to best practice standards and variability in interpretation of assessments across reviews, it was difficult for us to both provide a summary of the risk of bias or quality concerns of the evidence and to interpret this for readers.

We have identified numerous gaps in the current body of systemic reviews assessing clinical outcomes of prolonged infusions of beta-lactams. For one, the impact of concomitant non-beta-lactam antimicrobial therapy was not addressed consistently across reviews. Secondly, differences in total daily dose between infusion strategies was not regularly addressed. Studies originally touted prolonged infusion protocols as a potential cost saving mechanism, assuming similar efficacy of reduced dose prolonged infusion to full dose intermittent infusions of beta-lactams. However, given generally well tolerated beta-lactam antimicrobial therapies and the goal of improved clinical outcomes, it is imperative that systematic reviews assess the effect of prolonged infusions with consistent total daily doses. Finally, although beta-lactams doses and frequencies are regularly adjusted for patient specific PK/PD variations, such as impaired renal function and extremes of body weight, no review attempted to account for this or commented. We recognize that the gaps identified in systematic reviews may not reflect corresponding knowledge gaps in primary studies. However, systematic reviews should too describe gaps in included studies to guide future randomized control studies.

Our overview is primarily limited by the quality of included systematic reviews. Quality of included systematic reviews, as assessed with AMSTAR-2, was generally low given common critical deficiencies (i.e. lack of predetermined protocol and well-defined search strategies). Risk of bias, as assessed with the ROBIS tool, often highlighted the same deficiencies given overlapping criteria. Given that methodologic deficiencies were generally common to included reviews, we did not adjust or exclude reviews of lower quality or higher risk of bias. Rather, we have highlighted these to facilitate appropriate interpretation of findings. To capture all relevant reviews, we broadly searched multiple databases, with a peer reviewed search strategy developed by an experience information specialist. Despite this rigorous process, included systematic reviews may have systematically omitted primary studies introducing a selection or publication bias in the findings of our overview. Due to the wide variety of tools used to assess risk of bias and a lack of risk of bias assessment at the outcome level we were unable to synthesize this data in a meaningful way, and resource restrictions precluded us from taking the additional step of re-assessing all primary studies using appropriate tools for the study design. Discrepancies and variations in constructs in the risk of bias assessments make it difficult to truly assess risk of bias for the primary studies. Although overlap, as measured by CCA, was generally moderate to high for each clinical outcome studied, percentage of unique references ranged from 38–78%. Thus, as is typical of overviews, our findings are more strongly impacted by primary studies found in multiple reviews than not.

## Conclusion

This overview of reviews aimed to map and compare of objectives, methods, and findings of existing systematic reviews investigating prolonged beta-lactam infusions. A systematic and comprehensive review of the literature revealed a wealth of literature, both systematic reviews and primary studies. Despite this, conclusions across reviews on the same topic were inconsistent and potentially biased. Findings from our overview clearly demonstrate a consistent and

reproducible lack of harm with prolonged infusions. However, when qualifying and quantifying benefit of prolonged infusions of beta-lactams there is variability in effect size and significance of benefits of prolonged infusions. Findings of reduced mortality and improved rates of clinical cure should be tempered by presumed risk of bias. Despite 21 systematic reviews addressing prolonged infusions of beta-lactams, this overview supports the need for better conducted reviews and continued need for definitive trials given variability in scope of the available systematic reviews. Subsequent systematic reviews should only include well designed RCTs and should specifically evaluate the proposed benefits found in various subgroup-analyses—i.e. high risk of mortality. Given the generally low quality and variable risk of bias of included reviews, future systematic review should employ rigorous methods in keeping with AMSTAR-2 and ROBIS recommendations.

## Supporting information

**S1 Table. AMSTAR results of included reviews.** AMSTAR criterion: 1 –Did the research question and inclusion criteria for the review include the components of PICO?; 2 –Did the report of the review contain an explicit statement that the review methods were established prior to the conduct of the review and did the report justify any significant deviations from the protocol?; 3 –Did the review authors explain their selection of the study designs for inclusion in the review?; 4 –Did the review authors use a comprehensive literature search strategy?; 5 –Did the review authors perform study selection in duplicate?; 6 –Did the review authors perform data extraction in duplication?; 7 –Did the review authors provide a list of excluded studies and justify the exclusions?; 8 –Did the review authors describe the included studies in adequate detail?; 9 –Did the review authors use a satisfactory technique for assessing ROB in individual studies that were included in the review? 9a –RCT, 9b –NRSI; 10 –Did the review authors report on the sources of funding for the studies included in the review?; 11 –If meta-analysis was performed did the review authors use appropriate methods for statistical combination of results? 11a –RCT; 11b –NRSA; 12 –If meta-analysis was performed, did the review authors assess the potential impact of ROB in individual studies on the results of the meta-analysis or other evidence synthesis?; 13 –Did the review authors account for ROB in individual studies when interpreting/discussing the results of the review?; 14 –Did the review authors provide a satisfactory explanation for, and discussion of, any heterogeneity observed in the results of the review?; 15 –If they performed quantitative synthesis did the review authors carry out an adequate investigation of publication bias (Small study bias) and discuss its likely impact on the results of the review?; 16 –Did the review authors report any potential sources of conflict of interest, including any funding they received for conducting the review? Abbreviations: Y—Yes; PY—Partial Yes; N—No; NA—Not Applicable; CL—critically low quality; L—low quality; M—moderate quality. (DOCX)

**S2 Table. Risk of bias and/or quality assessments of primary studies as reported in included reviews.** NOS—Newcastle Ottawa Scale; ROB—Risk of bias. (DOCX)

**S3 Table. Characteristics of reviews reporting mortality.** [a] Subsequent sensitivity analyses conducted to separate randomized and non-randomized trials, PI—prolonged infusion, II—intermittent infusion, AMSTAR-2 –assessing the methodologic quality of systematic reviews, ROBIS—risk of bias tool for systematic reviews. (DOCX)

**S4 Table. Characteristics of reviews reporting clinical cure or treatment failure.** [a] Subsequent sensitivity analyses conducted to separate randomized and non-randomized trials, PI—

prolonged infusion, II—intermittent infusion, AMSTAR-2 –assessing the methodologic quality of systematic reviews, ROBIS—risk of bias tool for systematic reviews.
(DOCX)

**S5 Table. Characteristics of reviews reporting microbiologic cure.** PI—prolonged infusion, II—intermittent infusion, AMSTAR-2 –assessing the methodologic quality of systematic reviews, ROBIS—risk of bias tool for systematic reviews.
(DOCX)

**S6 Table. Characteristics of reviews reporting length of stay.** [a] Subsequent sensitivity analyses conducted to separate randomized and non-randomized trials, CI—Continuous infusion, PI—prolonged infusion, II—intermittent infusion, AMSTAR-2 –assessing the methodologic quality of systematic reviews, ROBIS—risk of bias tool for systematic reviews.
(DOCX)

**S7 Table. Characteristics of reviews reporting adverse events.** CI—Continuous infusion, PI—prolonged infusion, II—intermittent infusion, AMSTAR-2 –assessing the methodologic quality of systematic reviews, ROBIS—risk of bias tool for systematic reviews.
(DOCX)

**S8 Table. Characteristics of reviews reporting cost.** [a] Subsequent sensitivity analyses conducted to separate randomized and non-randomized trials, PI—prolonged infusion, II—intermittent infusion, AMSTAR-2 –assessing the methodologic quality of systematic reviews, ROBIS—risk of bias tool for systematic reviews.
(DOCX)

**S9 Table. Characteristics of reviews reporting emergence of resistance.** PI—prolonged infusion, II—intermittent infusion, AMSTAR-2—assessing the methodologic quality of systematic reviews, ROBIS—risk of bias tool for systematic reviews.
(DOCX)

**S10 Table. Characteristics of reviews reporting PK/PD outcomes.** PI—prolonged infusion, II—intermittent infusion, AMSTAR-2 –assessing the methodologic quality of systematic reviews, ROBIS—risk of bias tool for systematic reviews.
(DOCX)

**S1 Fig. Search strategy.**
(DOCX)

**S2 Fig. PRIO-harms checklist.**
(DOCX)

## Acknowledgments

We would like to acknowledge Becky Skidmore, MLIS and Lindsey Sikora, MLIS for developing, conducting, and reviewing the search strategy. We would also like to acknowledge Carole Lunny, MPH, PhD for reviewing the manuscript prior to publication.

## Author Contributions

**Conceptualization:** Pierre Thabet, Erika MacDonald, Salmaan Kanji.

**Data curation:** Pierre Thabet, Anchal Joshi, Salmaan Kanji.

**Formal analysis:** Pierre Thabet, Salmaan Kanji.

**Methodology:** Pierre Thabet, Erika MacDonald, Brian Hutton, Wei Cheng, Adrienne Stevens, Salmaan Kanji.

**Project administration:** Pierre Thabet, Salmaan Kanji.

**Resources:** Pierre Thabet.

**Supervision:** Salmaan Kanji.

**Visualization:** Adrienne Stevens, Salmaan Kanji.

**Writing – original draft:** Pierre Thabet, Salmaan Kanji.

**Writing – review & editing:** Pierre Thabet, Anchal Joshi, Erika MacDonald, Brian Hutton, Wei Cheng, Adrienne Stevens.

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
