## [Decision Letter · Decision Letter 0]

29 Sep 2020

PONE-D-20-13456

Clinical and pharmacokinetic/dynamic outcomes of prolonged infusions of beta-lactam antimicrobials: an overview of systematic reviews

PLOS ONE

Dear Dr. Kanji,

Thank you for submitting your manuscript to PLOS ONE. After careful consideration, we feel that it has merit but does not fully meet PLOS ONE’s publication criteria as it currently stands. Therefore, we invite you to submit a revised version of the manuscript that addresses the points raised during the review process.

ACADEMIC EDITOR:

Dear authors:

Your manuscript has been reviewed by two experts in the field, and they have found some points that need to be addressed before this manuscript is considered for publication. Please go through the reviewers' comments and consider addressing these points, and prepare a revised version.

I suggest special emphasis on the following major points:

1. The methodological issues raised by reviewer 2 are not minor and need to be addressed, otherwise the credibility of this work will be threatened.  (Jadas scale use, lack of appropriate question, the use of both AMSTRA and ROBIS justification a clearly out of date search)

2. Also, the relevance of this study was questioned by reviewer 1: "My question to the authors is what is new and so important with their findings and review of reviews then? This needs to be clarified and described clearly as presently, this study despite using a slightly different methodology is also reporting and describing similar results with existing studies". You should respond to this comment and provide a very strong rationale to justify this study considering the evidence that is already out there. Was there a real need for conducting an overview?

We look forward to receiving your revised manuscript.

Kind regards,

Ivan D. Florez

Academic Editor

PLOS ONE

Journal Requirements:

2. It appears there are two cases of broken references, one at the beginning of the results section, and one in the section which begins: "Nine of 14 reviews (21, 22, 26-29, 33, 45, 50) show lower mortality with prolonged infusions...". There are also some comments which I think are left over from drafting your paper ("Also would suggest you define what the ‘N’ and ‘u’ are referring to in the caption. I would also indicate in one sentence what the pattern shown is saying about these reviews to spell out for readers.") in the Findings, Mortality section. Please can you fix or remove these in your revised manuscript.

4. Please include a caption for figure 1.

Additional Editor Comments (if provided):

Dear authors:

Your manuscript has been reviewed by two experts in the field, and they have found some points that need to be addressed before this manuscript is considered for publication. Please go through the reviewers' comments and consider addressing these points, and prepare a revised version.

I suggest special emphasis on the following major points:

1. The methodological issues raised by reviewer 2 are not minor and need to be addressed, otherwise the credibility of this work will be threatened. (Jadas scale use, lack of appropriate question, the use of both AMSTRA and ROBIS justification a clearly out of date search)

2. Also, the relevance of this study was questioned by reviewer 1: "My question to the authors is what is new and so important with their findings and review of reviews then? This needs to be clarified and described clearly as presently, this study despite using a slightly different methodology is also reporting and describing similar results with existing studies". You should respond to this comment and provide a very strong rationale to justify this study considering the evidence that is already out there. Was there a real need for conducting an overview?

Reviewers' comments:

Reviewer's Responses to Questions

**Comments to the Author**

1. Is the manuscript technically sound, and do the data support the conclusions?

Reviewer #1: Partly

Reviewer #2: Partly

2. Has the statistical analysis been performed appropriately and rigorously? 

Reviewer #1: Yes

Reviewer #2: N/A

3. Have the authors made all data underlying the findings in their manuscript fully available?

Reviewer #1: Yes

Reviewer #2: Yes

4. Is the manuscript presented in an intelligible fashion and written in standard English?

Reviewer #1: Yes

Reviewer #2: Yes

5. Review Comments to the Author

Reviewer #1: Thank you for the opportunity to review this manuscript. Please find some of my comments and suggestions to improve the manuscript.

1. Line 61 – 62: agree but please include references to support this.

2. Line 63: such as drug clearance, metabolism and volume of distribution – felt like an incomplete and a hanging statement? Maybe the authors should add differences after i.e. …volume of distribution differences.

3. 40 – 50% fT>MIC: is this true for cephalosporins, where 60-70% fT>MIC was associated with better outcomes including in animal models and human studies. Clarifications/amendments are needed.

4. 100% fT>MIC: please add better references to support this e.g. McKinnon 2008 etc. References 11 and 12 didn’t identify this target.

5. Line 79: intravenous

6. Line 79 – 80: minor – I don’t really agree with prolonged infusion changing PK, PK remains the same e.g. Vd and CL doesn’t change with method of administration but PK/PD target attainment does change. I suggest modifying this.

7. Discussion: “these differences preclude us from broadly generalizing the benefits of prolonged infusions of beta-lactam antimicrobials” – this notion is already known and has been described in many reviews and studies previously i.e. prolonged or continuous infusion won’t benefit all patients but only a sub-group of patients who are severely ill. My question to the authors is what is new and so important with their findings and review of reviews then? This needs to be clarified and described clearly as presently, this study despite using a slightly different methodology is also reporting and describing similar results with existing studies.

8. Most review combined EI and CI so hard to comment between the two strategies – the authors could comment on PK/PD target attainment between the two?

9. Studies form different countries/continents can also be highlighted. E.g. the ones that would benefit more from prolonged infusion are those from countries with less susceptible pathogens vs. Europe and North America.

Reviewer #2: Thank you for the opportunity to review this manuscript. Although the authors have attempted to conduct an overview of reviews of beta-lactam infusions for treating infections, there are a number of methodological and reporting issues that seriously limit the credibility of the work. I have provided some specific suggestions that I think might help improve upon the current version:

Abstract:

-I think it is helpful in the objective to report the condition(s) being investigated as well as the population of interest, i.e., beta-lactam infusions for which condition(s) and for whom? In this case I think it would be hospitalized patients (adults?) with infections (any infection? sepsis or septic shock?).

Introduction:

-Line 56: Be careful to keep the verb tense consistent (as much as possible) throughout the main text. This sentence now uses past tense - do you mean 'has increased at a staggering pace'?

-You may consider shortening the introduction for readability, but otherwise it provides a nice background and good rationale for the work.

Methods:

-I suggest justifying why the Jadad guidance was selected as a framework for dealing with discordant systematic reviews, when modern guidance exists that is tailored specifically toward the conduct of overviews of reviews (the Jadad framework references decision making, but being nearly 25 years old is not tailored to overviews of reviews).

-I suggest justifying why an overview of reviews was the most appropriate methodology for addressing the research question. With just over 60 available trials (as mentioned in the Introduction), why was a systematic review not feasible? Important considerations (as outlined by Cochrane) include whether existing systematic reviews are up-to-date, whether they are sufficiently homogeneous with respect to populations, comparators, and/or outcome measures, and whether existing systematic reviews are of sufficiently low risk of bias (or high methodological quality). The description of available systematic reviews in the Introduction (lines 84-86) is not entirely convincing, so could perhaps be edited to provide a stronger rationale.

-I suggest reporting specifically which grey literature sources were searched and the search terms/strategy used.

-If feasible, consider running a search update since the searches are now >1 year old. If this is not feasible due to time and/or resource constraints, you may wish to include this as a limitation.

-It would be helpful to the reader if somewhere within the main text (ideally the introduction or beginning of the methods) you included an explicit, well-defined research question, including the population, intervention(s), comparator(s), outcome(s), length of follow-up, and setting(s) of interest for the overview of reviews. At present, it is not entirely clear.

-I suggest providing a more detailed description of the methods used to deal with missing, flawed, or discordant risk of bias assessments for the same studies across systematic reviews. Similarly, it would be helpful to report in more detail how you dealt with discrepant data from primary studies within the included systematic reviews during data extraction.

-As AMSTAR-2 and ROBIS include many overlapping items (despite aiming to measure different constructs, empirical evidence shows that the overall appraisals tend to be correlated), it would be helpful to justify why both were used instead of one or the other (which would have been sufficient).

-For AMSTAR 2, it would be helpful to define which domains you considered to be "critical" in the context of this overview of reviews.

Results:

-I suggest adopting consistent terminology throughout the main text to avoid confusion. For example, in line 150 you note that 21 studies were included when in fact (I believe?) these were unique systematic reviews. It will be important throughout the text to differentiate between the systematic reviews, and the primary studies included in them.

-Table 1: Just a detail, but technically AMSTAR 2 is used to appraise your confidence in the results of the systematic review (not 'quality' per se). I would suggest revising the title of the final column of the table to reflect this.

-Table 1: Since 'quality' or 'confidence' in review findings should (in theory?) be inversely related to risk of bias, you will need to explain how many of the systematic reviews appear to be of low or critically low quality, while also being at low risk of bias. Although I can appreciate that these are distinct constructs, as mentioned previously, the two tools are relatively similar in the concepts measured, so this will be important to justify. Also, how should readers interpret this discordance?

-Table 1: I would suggest reviewing the contents of the table for accuracy, especially in areas where "unspecified" has been used. For example, a cursory review of the systematic review by Roberts (2009) shows that the population of interest was hospitalized adults, while the table indicates "unspecified population".

-Under the interventions studied, it would be helpful to note, rather than simply "studied", which were the comparisons of interest (since these were all investigating comparative effectiveness?).

-Can you provide a more detailed description of overlap (i.e., for which outcome was overlap 5% vs. 78%)? This will be very important for the readers to be able to interpret the credibility of the results and conclusions.

-For review quality and risk of bias, please note that GRADE and the Cochrane RoB tool are measuring two separate constructs (GRADE measures outcome-level certainty of evidence, whereas the RoB tool measures study-level risk of bias). I suggest discussing these separately.

-Table 2 is not useful and should likely be deleted. Preferably, the information for all reviews would be included in an Appendix. Further, more work should be done to deal with the discordant assessments, rather than simply writing them off as 'discordant' (which is to be expected).

-Table 3 is not adding anything to what has already been described in Table 1, so I would suggest deleting it.

-It is not very helpful to provide ORs and RRs without 95% CIs. I would suggest revising to include these.

-Many of the reported results seem to focus on the high variability in the populations, interventions, outcomes, and their definitions, suggesting that perhaps an overview of reviews was not the ideal approach? You may wish to investigate this more carefully. Seems this was a severe limitation precluding strong conclusions?

-Again, Table 4 is not adding anything to what was already described in Table 1, so I would suggest deleting it.

-It would be helpful instead to present tables summarizing the findings (e.g., pooled estimates of effects) across the various systematic reviews.

-Figure 1 - I suggest here replacing "studies" with "systematic reviews".

6. PLOS authors have the option to publish the peer review history of their article (what does this mean?). If published, this will include your full peer review and any attached files.

Reviewer #1: No

Reviewer #2: **Yes: **Allison Gates

---

## [Author Response · Author response to Decision Letter 0]

23 Nov 2020

We have uploaded a response to reviewers including responses to the editors comments.

---

## [Decision Letter · Decision Letter 1]

21 Dec 2020

Clinical and pharmacokinetic/dynamic outcomes of prolonged infusions of beta-lactam antimicrobials:  an overview of systematic reviews

PONE-D-20-13456R1

Dear Dr. Kanji,

We’re pleased to inform you that your manuscript has been judged scientifically suitable for publication and will be formally accepted for publication once it meets all outstanding technical requirements.

Kind regards,

Ivan D. Florez

Academic Editor

PLOS ONE

Additional Editor Comments (optional):

The revised manuscript has been reviewed by one of the reviewers and myself as Associate Editor. We have concluded that you have addressed all the comments and the revised version has substantially improved to be considered for publication.

Reviewers' comments:

Reviewer's Responses to Questions

**Comments to the Author**

1. If the authors have adequately addressed your comments raised in a previous round of review and you feel that this manuscript is now acceptable for publication, you may indicate that here to bypass the “Comments to the Author” section, enter your conflict of interest statement in the “Confidential to Editor” section, and submit your "Accept" recommendation.

Reviewer #1: All comments have been addressed

2. Is the manuscript technically sound, and do the data support the conclusions?

Reviewer #1: Yes

3. Has the statistical analysis been performed appropriately and rigorously? 

Reviewer #1: Yes

4. Have the authors made all data underlying the findings in their manuscript fully available?

Reviewer #1: Yes

5. Is the manuscript presented in an intelligible fashion and written in standard English?

Reviewer #1: Yes

6. Review Comments to the Author

Reviewer #1: (No Response)

7. PLOS authors have the option to publish the peer review history of their article (what does this mean?). If published, this will include your full peer review and any attached files.

Reviewer #1: No

---

## [Editor Report · Acceptance letter]

2 Jan 2021

PONE-D-20-13456R1 

Clinical and pharmacokinetic/dynamic outcomes of prolonged infusions of beta-lactam antimicrobials: an overview of systematic reviews 

Dear Dr. Kanji:

I'm pleased to inform you that your manuscript has been deemed suitable for publication in PLOS ONE. Congratulations! Your manuscript is now with our production department. 

Kind regards, 

on behalf of

Dr. Ivan D. Florez; MD, MSc, PhD 

Academic Editor

PLOS ONE